# Closing the Performance Gap in Neural Conjugate Gradient Method: A Hybrid Multigrid Preconditioning Approach

## Abstract

Recent studies on neural preconditioners for the conjugate gradient (CG) methods have shown promise but sometimes over-optimistic: methods targeting small-scale problems are compared with decomposition-based preconditioners but do not scale; those aimed at moderate- to large-scale problems are typically $5\times$ slower in wall-clock time than state-of-the-art multigrid (MG) preconditioners, with worse iteration counts and higher model complexity. To address this gap, we revisit the designs of neural preconditioners and identify the key trade-off: methods tailored for scalability lack the expressiveness to emulate effective smoothers, whereas highly expressive designs struggle to scale. Building on these insights, we introduce a dual-channel neural multigrid preconditioner that couples a classical smoothing path with a lightweight neural convolutional path. This architecture preserves the minimal expressiveness and symmetric positive definite property while injecting data-driven adaptability. Our method demonstrates, for the first time, that neural preconditioners can surpass SOTA MG preconditioners on large-scale problems, achieving a 1.03-1.26$\times$ speedup on Poisson equations and 2-3$\times$ acceleration on other second-order PDEs involving up to 64 million unknowns, while also delivering 5-10$\times$ improvements over existing neural methods. These results establish a new benchmark for neural preconditioning.

## 1 Introduction

The conjugate gradient (CG) method (Nocedal & Wright, 2006) is an iterative algorithm for solving large-scale symmetric positive definite (SPD) linear systems, which frequently arise from the discretization of partial differential equations (PDEs) (Evans, 2022). It serves as a critical solver for many scientific and engineering applications, including linear elasticity, fluid dynamics, thermal simulation, and magnetostatics (Gould & Feng, 1994; Tagliafico et al., 2014; Harari & Hughes, 1991). Yet the convergence of CG strongly depends on the conditioning of the system, making preconditioning indispensable for efficiency.

Preconditioners act as approximators to the inverse operator, effectively reducing iteration counts of CG. Among classical approaches, multigrids (MG) (Briggs et al., 2000; McAdams et al., 2010; Tatebe, 1993; Shao et al., 2022) are regarded as state-of-the-art (SOTA) preconditioners in many PDE families on large-scale problems. By operating on a hierarchy of discretization levels, MG reduces error in different spatial frequencies and achieves rapid convergence with GPU-friendly implementations. Despite their strong performance, MG methods often require substantial domain-specific design and careful tuning of components such as smoothers and boundary handling to realize their full potential.

In recent years, neural approaches have emerged as an alternative pathway, aiming to learn preconditioners directly from data distributions rather than intensive domain knowledge. Neural preconditioners approximate inverse operators by data-driven learning, delivering adaptability and generalization across problem instances, where advantages have been widely shown on small-scale (up to 100K unknowns) problems over decomposition-based or weak MG preconditioners (Li et al., 2023; Rudikov et al., 2024; Han et al., 2024; Huang et al., 2022; Taghibakhshi et al., 2021; Luz et al., 2020a; Wang et al., 2023). However, comparisons have sometimes excluded the strongest MG base-

lines, such as AMGX (Naumov et al., 2015) in unstructured meshes and geometric MG (McAdams et al., 2010; Shao et al., 2022) in regular grids, leading to over-optimistic results. Our calibrated evaluations (Appendix B) show that SOTA neural preconditioners remain about five times slower in wall-clock time than SOTA MG methods on large-scale problems (8M unknowns), with worse iteration counts and higher model complexity. The inefficiency in both CG convergence rate and high model inference costs indicates two major limitations for prior works: (i) methods tailored for small-scale problems can be competitive with decomposition-based preconditioners but struggle to scale due to the heavy computational cost. For example, the per-variable evaluation of multilayer perceptrons (MLPs) in graph neural networks (Li et al., 2023), policy networks (Taghibakhshi et al., 2021), or neural operator layers (Rudikov et al., 2024) are too costly on large-scale problems compared with classical MG layers. (ii) scalable light-weighted designs, typically convolutional neural networks (CNN) (Kaneda et al., 2023; Han et al., 2024; Lan et al., 2024a), often lack the necessary expressiveness to emulate effective smoothing components of the MG method, making them less competitive in reducing errors in the MG framework, resulting in additional computing complexity in each iteration and with a poor CG convergence rate.

To address these challenges, we propose a dual-channel neural multigrid preconditioner that integrates a classical smoothing path with a lightweight neural convolutional path. The key idea is to preserve the well-established strengths of multigrid preconditioners, including its symmetric positive definite property, minimal expressiveness, and scalability, while injecting data-driven adaptability where classical methods are less effective. The convolutional channel is designed as a lightweight CNN-like operator that scales efficiently to tens of millions of unknowns, incurring negligible additional computational overhead compared to standard MG layers. At the same time, its learnable kernels provide sufficient expressiveness to complement classical smoothers, enabling more effective error reduction across different problem instances. By combining these two channels within the MG hierarchy, our design achieves a balance between robustness and adaptability: the classical path guarantees stability and efficiency, while the neural path captures problem-dependent structures that accelerate convergence.

To evaluate our approach and prior methods fairly, we propose a large and diverse dataset of linear second-order PDE systems, including Poisson, heat, and Helmholtz equations, involving up to 64 million unknowns with more than 3,000 instances. This dataset exceeds the scale of prior neural preconditioner benchmarks and matches the problem sizes where MG methods are typically deployed, establishing a rigorous testbed for neural and classical approaches.

Our experiments demonstrate that the proposed dual-channel neural MG preconditioner consistently outperforms both strong neural and MG baselines on large-scale problems. Specifically, it achieves a 1.03–1.26× speedup over geometric MG on Poisson equations, a 2–3× speedup on other second-order PDEs, and a 5–10× improvement over existing neural preconditioners. To our knowledge, this is the first demonstration that a neural preconditioner can surpass highly optimized MG methods on systems with tens of millions of variables.

In summary, our contributions are threefold:

- We provide a calibrated evaluation of existing neural preconditioners, showing that methods effective at small scales fail to scale and that scalable designs remain about $5\times$ slower than SOTA multigrid preconditioners on large problems;
- We construct a large-scale benchmark of linear SPD systems arising from second-order PDEs (up to 64M unknowns), including more diverse and challenging testcases;
- We propose a dual-channel neural multigrid preconditioner that integrates a classical smoothing path with a lightweight neural convolutional path, preserving the SPD property while improving both scalability and expressiveness.

We will open-source all of the code if the manuscript is accepted.

## 2 RELATED WORK

Neural approximators have been widely studied for solving PDEs under various geometric representations. Notable examples include neural operators (NOs) (Raonic et al., 2023; Li et al., 2020; Kovachki et al., 2023) and physics-informed neural networks (PINNs) (Cai et al., 2021), which have

| | Kaneda et al. (2023) | Han et al. (2024) | Lan et al. (2024a) | Ours |
|---|---|---|---|---|
| **Solver Type** | SD + Neural Prec. | Hybrid Iterative | SD + Neural Prec. | CG + Neural Prec. |
| **Maximized DoF** | 16M | 16M | 8M | 64M |
| **Accuracy Gain** | ✓ ($10^{-4}$) | ✓ ($10^{-4}$) | ✓ ($10^{-6}$) | ✓ ($10^{-6}$) |
| **Speedup over GMG** | × | × | × | ✓ |
| **Mixed B.C.** | × | × | ✓ | ✓ |
| **New PDE Types** | × (Poisson only) | ✓ (re-design) | × (Poisson only) | ✓ (re-train) |

Table 1: Comparison of SOTA accurate neural MG solvers for large-scale SPD linear equations on regular grids. ✓ indicates support, × indicates limitation. The solver type indicates the usage of the solver model, where "SD" stands for steepest descent and CG represents "conjugate gradient". "Mixed B.C." is short for the mixed Neumann and Dirichlet boundary condition Lan et al. (2024a) .

been applied across diverse PDE families. Domain-specific architectures such as MeshGraphNet (Pfaff et al., 2020) and FluidNet (Liu et al., 2016) also show that neural networks can approximate numerical schemes in particular physical systems. These approaches are often orders of magnitude faster than classical solvers, but their accuracy is frequently limited, with failures in long-term stability or generalization to unseen domains (Kaneda et al., 2023; Lan et al., 2024a). This motivates the integration of neural components into classical iterative methods, retaining their accuracy guarantees while leveraging data-driven adaptability to accelerate convergence.

Neural preconditioners have shown promise but remain largely constrained to small-scale problems or weak baselines. For instance, Li et al. (2023) proposed a GNN that predicts sparse decompositions and demonstrated improvements over decomposition-based preconditioners, but only up to 23K degrees of freedom (DoFs). Other lines of work extend neural operator approaches to the CG setting, including DeepONet-based preconditioners (Kopaničáková & Karniadakis, 2025) and FNO-CG methods (Rudikov et al., 2024). However, both GNN- and NO-based preconditioners incur significant computational costs, often requiring MLP reasoning over every unknown variable in each layer. This makes them prohibitively expensive for systems with tens of millions of degrees of freedom, where they lose competitiveness against strong multigrid (MG) methods. A broader class of works has explored combining MG with neural models (Xie et al., 2025; Cui et al., 2025; Antonietti et al., 2023; Kopaničáková & Karniadakis, 2025; Luz et al., 2020b; Huang et al., 2021), but most remain limited to small-scale settings. Reinforcement learning–based methods (Taghibakhshi et al., 2021) and decomposition-based neural preconditioners (Li et al., 2024) also report runtime costs that are one to two orders of magnitude slower than MG for million-DoF systems, underscoring the difficulty of scaling.

Multigrid methods represent one of the most effective families of numerical preconditioners for large-scale SPD systems, especially those arising from second-order PDEs such as Poisson's equation (Bolz et al., 2005; Pharr & Fernando, 2005; McAdams et al., 2010). The MG framework reduces errors across spatial frequencies via coarsening strategies that build hierarchies of discretizations (Tatebe, 1993; Briggs et al., 2000). Modern implementations such as geometric MG (GMG) (McAdams et al., 2010; Shao et al., 2022) and algebraic MG (AMG) (Naumov et al., 2015; Demidov, 2019) are widely recognized for their efficiency and scalability. Recent neural MG approaches aim to extend these ideas: UGrid (Han et al., 2024) replaces components of the canonical V-cycle with a learned linear model on 2D grids, while MLPCG (Lan et al., 2024a) achieves SOTA neural preconditioning on 3D Poisson systems up to $256^3$ through a customized neural smoother AffConv layer, outperforming AMGs (Demidov, 2019; Naumov et al., 2015) and other neural baselines such as DCDM (Kaneda et al., 2023) and FluidNet (Sundaresan et al., 2013). We take MLPCG as the SOTA neural baseline and compare against it extensively in our experiments.

## 3 BACKGROUND

The multigrid (MG) method is generally recognized as one of the most competitive high-performance solvers for large-scale linear systems. MG solvers are categorized into geometric

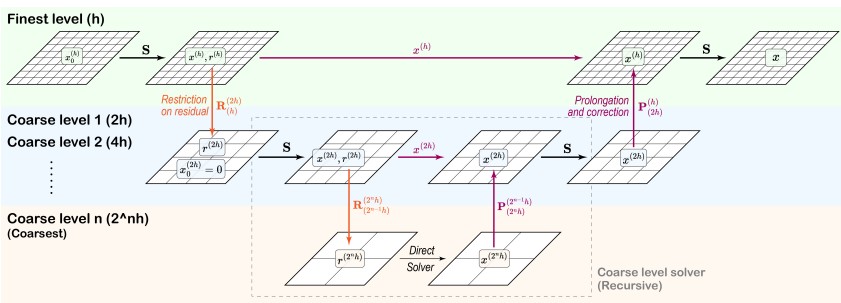

Figure 1: The Multigrid V-cycle scheme.

multigrid (GMG) and algebraic multigrid (AMG), and the GMG method is considered one of the SOTA methods for problems with a geometric structure, e.g., PDEs discretized on regular grids.

The main idea of GMG is to construct a hierarchy of grids with decreasing resolution, and to carry out the following linear operations on grids to pass and update fields stored in hierarchical grid (Fig. 4): **Restriction $R$**: A down-sampling matrix that maps a field on a high-resolution grid to a low-resolution grid. On the structured grid, it can be chosen as the average pooling operator; **Prolongation $P$**: An up-sampling matrix that maps a field on a low-resolution grid to a high-resolution grid. Often we choose $P = R^\top$ for the sake of symmetry; **Smoothing $S$**: An iterative solver that refines intermediate solutions of the equation on the current grid, which is typically chosen from basic and highly-parallel GPU solvers like Jacobi or Gauss-Seidel iterations. The smoothing operation typically involves two linear operators $H$ and $S$ depending on the type of the smoother and the matrix $A$, which updates the solution field $x$ as $x \leftarrow Sb + Hx$.

Starting from an input equation $Ax = b$, we define a grid hierarchy named $(h), (2h), (4h), ...$ from the finest grid (identical to the original domain) to the coarsest grid following the naming convention. GMG first performs a pre-smoothing operation on the initial guess $x_0^{(h)}$ and produces a refined solution $x^{(h)}$. Then a restriction operation (e.g. the average pooling) is applied to the residual vector to produce the right-hand-side vector on the coarser grid $b^{(2h)} = R_{(h)}^{(2h)} r^{(h)} = R_{(h)}^{(2h)}(b - Ax^{(h)})$.

After this, the GMG method solves the residual on a coarser grid. We may set the initial guess $x_0^{(2h)} = 0$ and thus the whole smooth-restriction routine could be recursively applied until the coarsest level. At the bottom level, the problem scale is small enough to be solved via precise direct solvers. After solving the coarse problem, the solution is prolongated to the finer level.

Tracing back the recursion, after receiving the prolongated solution from the coarser level, we update the solution $x^{(h)} = x^{(h)} + P_{(2h)}^{(h)} x^{(2h)}$, followed by another smoothing operation to produce the final solution. The solution is then up-sampled until we reach the finest grid and terminate the algorithm.

The whole procedure above is called a V-cycle, which is a typical and widely used GMG scheme. It outputs a refined approximation to the solution $x = A^{-1}b$ that is much more precise than the initial guess $x_0$. The power of GMG is to apply smoothing on grids with different resolutions, thus effectively tackling the residual error with different spatial frequencies.

The complicated V-Cycle scheme can have numerous variants by adjusting the choices of the three linear operators $R, P, S$. Its performance could also fluctuate greatly depending on how much the choices "match" the problems. Poor matches may lead to degraded performance, stagnation, or even failure to converge (Briggs et al., 2000). To improve the convergence rate and robustness of the multigrid method, the V-cycle scheme is usually incorporated into the preconditioned conjugate gradient (PCG) method, known as MGPCG (Tatebe, 1993). As an iterative solver, the number of iterations of PCG depends on the quality of its preconditioner, and the cost of each single step depends on the complexity of solving the preconditioner. The property of quick approximation makes GMG suitable for being a preconditioner in PCG. The whole V-cycle model, viewed as an operator, is proved to be linear and SPD under certain types of smoothers (Tatebe, 1993).

## 4 MODEL

### 4.1 MOTIVATION

As noted by previous works of neural preconditioners (Lan et al., 2024a; Han et al., 2024), the V-cycle of a geometric multigrid is analogous to a single-channel linear CNN with non-learnable kernels excluding bias terms or activation functions. A neural CNN model can be trained to be a good approximator, offering a promising inspiration to augment the MG framework with learning methods. However, it remains a challenge to design a preconditioner that satisfies the numerical requirements for a convergence guarantee of the PCG method. When introducing bias terms, activation functions, or arbitrarily learnable kernels, the preconditioner violates the requirements of the SPD property, invalidating the PCG convergence guarantee. Previous methods resorted to steepest descend (Kaneda et al., 2023; Lan et al., 2024a) or mixed iterative methods (Kaneda et al., 2023; Kopaničáková & Karniadakis, 2025) that does not fully exploit the numerical property of the V-cycle scheme, leading to suboptimal performance and additional cost (e.g. the projection step introrduced in Lan et al. (2024a)).

In contrast, we propose a learnable convolutional kernel in the neural preconditioner, which corresponds to a smoother in the MG-like model. However, the requirements of SPD strongly constrains the design of an arbitrarily parameterized smoother. As a necessary condition, it requires the same convolutional operation on different layers, which highly limits the expressiveness. Therefore, we carefully examine an alternative but mathematically equivalent V-cycle model, leading to a flexible parameterization while preserving the SPD property.

### 4.2 V-CYCLE IN DUAL CHANNELS

It is not immediately obvious why a single-channel CNN-like structure constitutes an SPD linear operator without delving into a formal proof. An equivalent expression of the V-cycle operator, as shown in Tatebe (1993), clarifies the connection:

$$\boldsymbol{V}^{(l)} = \boldsymbol{S}_{2k} + \boldsymbol{H}^k \boldsymbol{P} \boldsymbol{V}^{(l-1)} \boldsymbol{P}^\top \boldsymbol{H}^{\top k} \tag{1}$$

where $\boldsymbol{V}^{(l)}$ is the V-cycle operator at level $l$ with $\boldsymbol{V}^{(0)} = \boldsymbol{A}^{(0)\,-1}$, and $\boldsymbol{R} = \boldsymbol{P}^\top, \boldsymbol{P}$ refer to the restriction and prolongation. The $\boldsymbol{S}_{2k}$ and $\boldsymbol{H}$ correspond to the linear operators in the smoother, where $k$ is the number of smoothing iterations. We use $\boldsymbol{S}_{2k}$ to denote the resulting matrix applied on $\boldsymbol{b}$ that consists of both pre- and post-smoothing. The SPD property of $\boldsymbol{V}^{(l)}$ can then be proved by induction on the level, with the SPD property of $\boldsymbol{V}^{(0)}$ and $\boldsymbol{S}_{2k}$. This expression suggests an alternative but mathematically equivalent formulation of the V-cycle model: a dual-channel version, where one channel computes $\boldsymbol{S}_{2k}$ (S-channel) and the other computes $\boldsymbol{H}^k \boldsymbol{P} \boldsymbol{V}^{(l-1)} \boldsymbol{P}^\top \boldsymbol{H}^{\top k}$ (H-channel).

In this formulation, the S-channel performs smoothing directly on the right-hand-side vector without down-sampling to a coarser resolution. In contrast, the H-channel first interprets the right-hand-side vector $\boldsymbol{b}$ as the initial guess $\boldsymbol{x}_0$, applies the transposed post-smoothing operator, then down-samples the result to the next level without computing the residual explicitly, and finally prolongates back with the post-smoother.

Our ablation studies confirm that this dual-channel model exhibits identical convergence behavior to the original V-cycle scheme, with negligible computational overhead when implemented appropriately (Appendix H).

### 4.3 OUR MODEL DESIGN

The above dual-channel split implies a natural design of a neural model that can learn better smoothers from data priors: the H-channel can be replaced by a learned linear CNN module $\boldsymbol{H}(\theta)$, while the S-channel has to be kept. As long as the transposed operator of $\boldsymbol{H}(\theta)$ is applied afterwards, the overall preconditioner remains SPD throughout the training process.

Our model design therefore follows the dual-channel V-cycle model, but replaces the $\boldsymbol{H}$ operator on each level with customized linear neural layers. Intuitively, the neural convolutional layer is

interpreted as a neural smoother in the multigrid context. The optimal smoother for solving $\boldsymbol{Ax} = \boldsymbol{b}$ is intuitively $\boldsymbol{A}^{-1}$, which is linear in operation but non-linear with respect to $\boldsymbol{A}$. This observation inspires us to design a spatially variant convolutional operator, whose kernel is non-linear with respect to stencils of $\boldsymbol{A}$.

Previous works have shown the power of the vanilla convolutional layers (Kaneda et al., 2023; Han et al., 2024) on spatially invariant Poisson equations with uniform boundary conditions. However, they struggle in the case with mixed boundary conditions or spatially varying PDEs since they do not capture the spatially variant information of the matrix $\boldsymbol{A}$. The AffConv layer introduced by Lan et al. (2024a) tries to address this issue by introducing an affine layer with respect to local voxel information to adapt to mixed boundary cases, but is largely linear with respect to the local stencil of $\boldsymbol{A}$, resulting in limited performance and cannot generalize to the spatially varying PDEs.

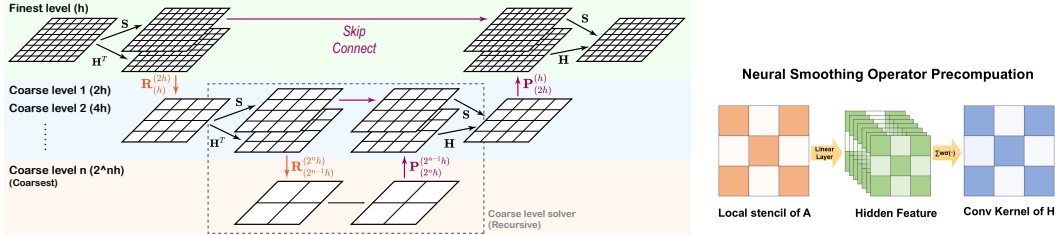

Figure 2: The overview of our model. We follow the dual-channel V-cycle model, but replace the smoothing operator in one channel by a precomputed neural smoother.

We introduce a spatially varying equivariant feed forward convolution layer, which is a two-layer feed-forward neural network that maps to a local stencil to the kernel of the neural smoother. Fig. 2 We also consider the equivariancy with respect to rotation or axis swapping, with some DoFs of the linear mapping factorized out. The spatially varying convolutional operators are precomputed before the PCG iteration.

We conclude the property of our model design: (i) preserves the SPD property throughout training; (ii) inherits the fast convergence rates of the geometric multigrid scheme by setting the initial guess identical to the V-cycle scheme; (iii) provides a moderate-scale parameterization space; (iv) has a comparable inference time to a classical geometric multigrid method.

## 5 TRAINING

### 5.1 DATASET

We conducted a large-scale dataset that contains three families of linear second-order PDEs: Poisson's equations, Heat equations, and Helmholtz equations. In each PDE family, we generated more than 200 instances by varying boundary conditions and physical parameters. The finite difference method was used to discretize these PDEs on 3D grid with a scale ranging from $256^3$ to $512^3$, resulting in large-scale linear systems (e.g, $Ax = b$). We implemented a large-scale linear system solver with a high-performance GPU programming language, Taichi, to solve these equations and generated the ground-truth training and testing dataset. Fig. 3 shows some of our generated PDEs. For each family of PDE, we took $1/30$ for training and the rest for testing. Below, we elaborate on the details of the three families of PDEs.

**Poisson's equation in Fluid Animation**  Fluid dynamics governed by the Navier-Stokes equation can be solved by a classic stable-fluid method Stam (2023); Bridson & Müller-Fischer (2007), typically bottlenecked by its Poisson equation solving at each frame. We follow the line of works Kaneda et al. (2023); Lan et al. (2024a) and construct a family of Poisson equation that solve the pressure field in fluid simulation in six carefully designed scenes, with a grid resolution from $256^3$ to $512^3$. We not only include mixed Dirichlet and Neumann boundary conditions, but also include more diverse object movements including translation, rotation, and deformation, incurring more turbulence in fluid motion. The dataset collects about 2K Poisson's equation instances from all six

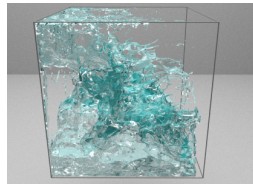
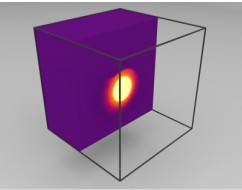
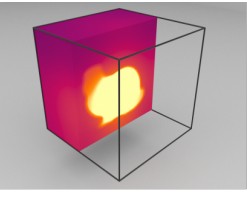
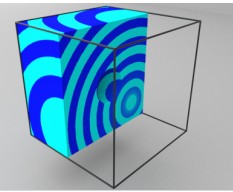

Poisson's equation        Heat equation        Helmholtz equation

Figure 3: Gallery of some of the generated PDEs. Leftmost: key frames of the fluid animation dataset for Poisson equations. Middle two: the heat transferring simulation dataset for heat equations. Rightmost: The synthetic Helmholtz equations. Please refer to our videos in the Supplemental Material for more details.

scenes, each scene includes unique $A$ and by temporally varying boundary conditions and fluid-solid distribution. We refer readers to our supplementary video for the visualization of these scenes.

**Heat Transfer** We consider the parabolic PDE arising in the heat transfer procedure. After the temporal and spatial discretization, the equation solved in each frame is in the form $\nabla \cdot (\beta(\boldsymbol{x})\nabla \boldsymbol{u}) = \frac{\partial \boldsymbol{u}}{\partial t}$, where $\boldsymbol{u}$ is the unknown field to solve, and $\beta$ is a spatially-varying coefficient field determined by the thermal conductivity. There are 300 instances with a grid resolution of $256^3$ in this kind of PDE, $1/30$ of them were selected for training, and the rest for testing.

**Helmholtz equation** We consider the Helmholtz equation, a type of elliptic PDE widely used in the field of optics and electromagnetism. The PDE is expressed as $\Delta \boldsymbol{u} + k(\boldsymbol{x})^2 \boldsymbol{u} = \mathbf{Q}$, where $k$ is a spatially varying wave number, and $\mathbf{Q}$ is the source term. We conducted temporally changed source term and boundary conditions to generate 300 unique instances with a grid resolution of $256^3$. We also picked $1/30$ of the instances for training and the rest for testing.

### 5.2 LOSS AND OPTIMIZER

To efficiently train a preconditioner with a good loss design remains a challenge in learning neural preconditioners. Matrix-level loss functions, for example, the condition number or the distance with the inverse of the matrix, are unavailable for large-scale linear systems due to their computational inefficiency. Li et al. (2023) discusses about using either $\|\boldsymbol{A}\boldsymbol{M}(\boldsymbol{b}) - \boldsymbol{b}\|_2$ or $\|\boldsymbol{M}(\boldsymbol{A}\boldsymbol{x}) - \boldsymbol{x}\|_2$ as the supervision signal, but such ideas are reported to be ineffective in Kaneda et al. (2023); Demidov (2019). Kaneda et al. (2023); Demidov (2019) improve the loss by adding more right-hand-side vectors from the Ritz decompositions of the matrix. However, such decomposition requires unacceptable time consumption for large-scale problems considered in our setting. We also observe that these losses are misaligned with the CG performances of the trained preconditioner. (Sec. H.2)

We empirically find the best choice of the loss function to be the residual after the $k$-th iteration of the conjugate gradient solver, a choice similar to Han et al. (2024). Since our model and loss design result in challenging computational graphs for auto-differentiation tools, instead we turn to a gradient-free optimizer, the CMA-ES optimizer Hansen et al. (2003) to train the neural preconditioner.

## 6 EXPERIMENTS

### 6.1 MOTIVATING EXAMPLE

Our experiments show that the SOTA neural methods targeting on large-scale Poisson equation solving (Kaneda et al., 2023; Han et al., 2024; Lan et al., 2024a) are often $5\times$ slower than the geometric MG preconditioners in their released dataset. The detailed evaluation is in Appendix B. The result reveals that the neural MG methods on large-scale problems are often more costly but still ineffective in reducing the error compared with the GMG model, indicating the insufficient effectiveness in their light-weight designs.

## 6.2 Experimental Setup

**Baselines**   We consider methods for solving the linear systems in the preconditioned conjugate gradient (PCG) frameworks of two categories of baselines, that is, MG preconditioners and neural preconditioners.

The numerical baselines include AMGX (Naumov et al., 2015) and AMGCL (Demidov, 2019), two widely-used open-source algebraic MG preconditioner implementations on GPUs. The AMGX provides 62 configurations in Naumov et al. (2015), and we have exhaustively searched through all of them and chosen the best one for a fair comparison. In addition, we implement a state-of-the-art geometric MGPCG method (GMG) (McAdams et al., 2010; Shao et al., 2022) in Taichi programming language (Hu et al., 2019), based on a line of third-party references (Sun, 2024; Hu et al., 2019; 2021) for a fair comparison.

We use MLPCG method (Lan et al., 2024a), the SOTA neural PCG method for Poisson's equation from fluid simulation on 3D grids, as our neural preconditioner baseline on Poisson problems. Other potential baselines include DCDM (Kaneda et al., 2023), UGrid (Han et al., 2024), and FluidNet (Sundaresan et al., 2013). However, these methods do not handle mixed Dirichlet and Neumann boundary conditions as our method and MLPCG do.

**Model Configuration**   We choose the Red-black Gauss-Seidel (Briggs et al., 2000) with smoothing iterations $k = 2$. The number of hidden channels (number of green intermediate kernels in the right subfigure of Fig. 2) is chosen to be $h = 8$ in our novel neural smoother to avoid heavy computational overhead. The number of multigrid levels (CNN hierarchies) is $m = 6$. The total number of parameters to be optimized are $\Theta(m \cdot k \cdot h)$.

**Performance metrics**   We consider two metrics for measuring a method's performance. Our main metric is the solving time for a method to solve the linear equation on a GPU. This time cost typically covers two stages: setting up a preconditioner (e.g., specific matrix analysis and factorization operation in AMGX and AMGCL) and running CG iterations. We measure the time consumption by running CG iterations until the relative residual error reaches a prespecified tolerance $\epsilon$, at which we consider the solving converges. We use $\epsilon =$ 1e-6 in Sec. 6.3, aligned with Lan et al. (2024a). For more involving metrics such as the CG residual curve, please refer to the Sec. H.4.

The training of our method on each type of PDE typically uses 10 hours on an NVIDIA A100 GPU, much faster than Lan et al. (2024a).

## 6.3 Comparisons to classical MG Preconditioners

Tbl. 2 and Tbl. 3 compare the average solving time of our methods with different numerical baselines. Note that the geometric multigrid baseline is substantially faster than AMGCL and AMGX because it exploits the geometric structure in regular grids with no sparse matrix-based implementation overheads. The over $15\times$ solving time difference between AMGCL, AMGX, and GMG underscores that the family of MG methods is highly sophisticated, exhibiting drastic performance variations before and after employing problem-specific designs. In particular, a fine-tuned MG carefully designed by experts with problem-specific knowledge is highly competitive, which we advocate should be a strong baseline more suitable than AMGCL or AMGX used in previous works.

Finally, our method is the fastest among all methods reported in Tbl. 2: our method is about $10\times$ faster than AMGCL and AMGX on the fluid Poisson dataset, and $1.5 - 15\times$ on other PDEs. The boost comes from exploiting the geometric structure in regular grids without sparse matrix-based implementations. Our method is $1.03$-$1.26\times$ faster than the strong geometric multigrid preconditioner due to the iteration reduction, induced by the neural channel exploring an expressive space of local operator tailored to the data prior.

## 6.4 Comparisons with Neural MG Preconditioners

We show the time profiling result on a typical piece of problem instance in our dataset, where our method achieves a $5\times$ speedup over MLPCG on Poisson datasets. Compared with MLPCG, our model outperforms in three dimensions: model inference time, costs on the iterative methods, and

Table 2: The time cost (in milliseconds) for solving linear equations with baselines and our method. Numbers in parentheses in the Dataset column are the size (in millions) of the linear problems. The Poisson* is obtained from the test problems used in MLPCG (Lan et al., 2024b). Speedup (MG) and Speedup (neural) report the performance boost of our method over the best MG baseline (GMG) and the SOTA neural baseline (MLPCG), respectively.

| Dataset | Scene | GMG | AMGCL | AMGX | MLPCG | Ours | Speedup (MG) | Speedup (neural) |
|---------|-------|-----|-------|------|-------|------|--------------|------------------|
| Poisson (16M) | Ball | 172 | 4378 | 721 | 1429 | **156** | 1.10× | 9.16× |
| | Spin | 185 | 4280 | 691 | 1670 | **179** | 1.03× | 9.33× |
| | Prop. | 167 | 4419 | 709 | 1448 | **156** | 1.07× | 9.28× |
| | Fish | 175 | 4228 | 750 | 1620 | **159** | 1.10× | 10.19× |
| | Robot | 171 | 4209 | 704 | 1558 | **157** | 1.09× | 9.92× |
| Poisson* (8M) | Ball | 135 | 2254 | 1578 | 724 | **115** | 1.17× | 6.30× |
| | Torus | 82 | Fail | 645 | 353 | **71** | 1.15× | 4.97× |
| Heat (16M) | Bunny | 641 | 5659 | 449 | N/A | **135** | 3.33× | N/A |
| | Syn. | 193 | 5939 | 284 | N/A | **88** | 2.19× | N/A |
| Helmholtz (16M) | Syn. | 502 | 3998 | 1150 | N/A | **182** | 2.76× | N/A |

Table 3: Time profiling comparison of baseline methods and our model on a typical problem instance selected in the "Propeller" scene of Poisson problem. The time is reported in milliseconds.

| Metric | GMG | AMGCL | AMGX | MLPCG | Ours |
|--------|-----|-------|------|-------|------|
| CG Iteration Count $n$ | 15 | 16 | 24 | 40 | 13 |
| Precomputation Time $t_{pre}$ | 0.2 | 3822 | 291 | 0 | 1.9 |
| Time per CG Iteration $t_{cg} = t_m + t_o$ | 11.6 | 26.0 | 19.7 | 28.1 | 12.0 |
|     Model Inference $t_m$ | - | - | - | 15.7 | 6.0 |
|     Other $t_o$ | - | - | - | 12.4 | 6.0 |
| Overall Time $t = t_{pre} + nt_{cg}$ | 175 | 4237 | 785 | 1124 | 158 |

convergence rate. MLPCG (Lan et al., 2024a) adopts a neural-preconditioned steepest descent with A-orthogonality (NSPDO) iterative methods, which introduces extra costs on the projection step due to the model asymmetry. Furthermore, our model is more lightweight and results in a shorter model inference time. Also, most importantly, our iteration count is substantially shorter than MLPCG, indicating a better numerical preconditioner.

## 7 CONCLUSION

We have calibrated the reporting bias in the field of neural preconditioning. Based on empirical insights, we have introduced a dual-channel neural multigrid preconditioner for the conjugate gradient (CG) method, closing the gap between the expressiveness of neural preconditioners and the scalability of classical multigrid approaches. By testing on a large-scale dataset with systems including up to 64 million unknowns, we have shown that the carefully tuned geometric MG preconditioners remain strong baselines, but our method surpasses them for the first time. Specifically, it achieves a 1.03–1.26× speedup over geometric MG on Poisson problems, 2–3× acceleration on other second-order PDEs, and 5–10× improvement over existing neural preconditioners. These results demonstrate that neural preconditioners can achieve both scalability and competitiveness at the problem sizes where MG is traditionally deployed.

Our method is not without limitations. Since it is built upon the PCG framework, it inherits its restriction to SPD systems and does not directly apply to indefinite or unsymmetric problems. Nevertheless, the proposed dual-channel design provides a promising blueprint for combining classical numerical methods with learning adaptability, and we believe it will inspire the development of neural methods for a broader class of iterative solvers in scientific and engineering computing.

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

# A    Discretization of PDE on Regular Grids

We consider a family of linear second-order PDEs that can be summarised as:

$$\nabla \cdot (\alpha(\boldsymbol{x})\nabla \boldsymbol{u}) + \beta(\boldsymbol{x})\boldsymbol{u} = \boldsymbol{f}, \quad \boldsymbol{x} \in \Omega,$$
$$s.t. \quad \boldsymbol{u}(\boldsymbol{x}) = g_D(\boldsymbol{x}), \quad \boldsymbol{x} \in \Gamma_D, \tag{2}$$
$$\nabla \boldsymbol{u}(\boldsymbol{x}) \cdot \boldsymbol{n}(\boldsymbol{x}) = g_N(\boldsymbol{x}), \quad \boldsymbol{x} \in \Gamma_N,$$

where $\Omega \subset \mathbb{R}^d$ is the problem domain and $d$ is the dimension of the problem. Our method primarily focuses on 3D problems but is straightforward to generalize it to solve 2D problems. $\boldsymbol{u} : \Omega \to \mathbb{R}$ is an unknown scalar field to be solved, $\boldsymbol{f} : \Omega \to \mathbb{R}$ is a source term, $\alpha(\boldsymbol{x})$ and $\beta(\boldsymbol{x})$ could be spatially varying coefficient or remain constant. $g_D$ and $g_N$ specify the Dirichlet and Neumann boundary conditions on $\Gamma_D$ and $\Gamma_N$, respectively, where $\Gamma_D \cup \Gamma_N = \partial\Omega$ partitions the domain boundary.

We consider discretizing a PDE on a regular grid with finite differences (Solomon, 2015), resulting in a linear system, $\boldsymbol{Ax} = \boldsymbol{b}$, where $\boldsymbol{A} \in \mathbb{R}^{n \times n}$ is a large-scale diagonal matrix, $\boldsymbol{x} \in \mathbb{R}^n$ is the unknown values stored at grid cells, and $\boldsymbol{b}$ is the right-hand-side that contains source term and boundary conditions. The matrix $\boldsymbol{A}$ is large and sparse since we use finite differences to discretize the operator and each unknown is only affected by its neighboring cells. $\boldsymbol{A}$ is also typically positive definite for elliptic PDE and some other types of PDE in practice. We handle Dirichlet and Neumann boundary conditions by modifying $\boldsymbol{A}$ and $\boldsymbol{b}$ as described in Bridson & Müller-Fischer (2007); Kaneda et al. (2023); Lan et al. (2024a).

# B    Motivating Example: Suboptimal Solutions in Previous Works

Despite rapid progress in neural PDE solvers and preconditioners, reported results are often over-optimistic due to comparisons to weak baselines, as shown in a well-known analysis (McGreivy & Hakim, 2024). We have confirmed the conclusion by a careful tests, showing that when calibrated fairly, neural preconditioners for the conjugate gradient (CG) method typically underperform by a significant margin: on moderate- to large-scale Poisson problems, they are often $5\times$ slower in wall-clock time than geometric MG baselines, while also requiring more CG iterations and incurring higher FLOP complexity per iteration. For details, please refer to the Poisson (8M) MLPCG test cases in Tbl. 2 and the UGrid test cases in Tbl. 4.

A major source of inefficiency is the computational overhead. Neural preconditioners usually spend additional time in applying complex network layers inside each CG iteration. In contrast, classical MG preconditioners build hierarchies quickly and apply smoothers with minimal cost, ensuring consistently fast convergence. Lightweight neural approaches on structured grids (e.g., CNN-based smoothers) are more efficient than graph or operator networks, but prior works often excluded comparisons against geometric MG preconditioners, with a single-channel CNN-like structure highly efficient in computation and without long precomputation building costs. Even with complicated boundary conditions or spatially varying coefficients, SOTA geometric MG remains highly competitive and difficult to surpass. Previous designs, such as uniform CNN kernels in Han et al. (2024); Kaneda et al. (2023), or the AffConv layer in Lan et al. (2024a), are hard to even emulate a classical Jacobi smoother, making them have more computational complexity (more channels / larger kernels) but less expressiveness in reducing the condition number of the system.

Neural methods targeting unstructured meshes face additional challenges. While their flexibility is attractive, their practical performance on large-scale problems remains limited. For example, GNN-based preconditioners with MLP computations in each message-passing layer (Li et al., 2023) cannot be trained at scale due to GPU memory constraints and are prohibitively slow at inference time. Similarly, reinforcement learning approaches (Taghibakhshi et al., 2021) and neural-operator-based preconditioners (Rudikov et al., 2024) also incur heavy computational costs that prevent effective scaling. In contrast, our case study shows that the AMGX implementation of a V-cycle MG-preconditioned CG method with weighted-Jacobi smoothers can solve Poisson problems on unstructured surface meshes with up to 2M degrees of freedom in under 100 ms to a relative error of $10^{-6}$ on an NVIDIA A100 server, consistently outperforming the neural methods.

These observations suggest that the key difficulty lies not in surpassing weak decomposition-based baselines, but in achieving competitiveness against strong multigrid preconditioners on large-scale

Table 4: Solving time on a Nvidia 4080 GPU and iteration used for problems in Han et al. (2024). Reported in time(ms)/iteration. The fixed error tolerance is used and set to be 1e-4 with respect to the initial residual of the linear system. Note, due to the rounding error, we cannot get a 1e-4 convergent result with 32-bit floating precision in XL and XXL scale in some test cases. We therefore run geometric multigrid (GMG) preconditioned CG in 64-bit floating precision in all XXL experiments and XL experiments with $*$. The result reveals that the hybrid iterative neural solver proposed in Han et al. (2024) is often more than $5\times$ slower than GMG on large-scale problems. The different inference time per iteration sources from the different number of DoFs for each instance.

| SIZE | XXL($4096^2$) | | XL($2048^2$) | |
| METHOD | UGRID | GMG | UGRID | GMG |
|---|---|---|---|---|
| BAG | 1573.1/116 | 118.4/11 | 41.6/24 | 24.7/10 |
| CAT | 974.4/72 | 125.6/9 | 39.2/20 | 23.4/9 |
| LOCK | 776.0/56 | 73.9/10 | 30.3/16 | 23.3/9 |
| N. INP. | 751.2/56 | 134.1/10 | 28.2/16 | 23.1/9 |
| NOTE | 164.0/12 | 66.1/9 | 21.4/12 | 22.7/9 |
| S. FEAT. | 995.3/72 | 192.7/8 | 40.1/20 | 26.9/8 |
| L-SHAPE | FAIL | 152.3/12 | FAIL | 54.6/11$^*$ |
| LAP. SQ. | 184.9/12 | 189.8/8 | 22.9/12 | 23.8/7 |
| P. SQ. | FAIL | 197.8/9 | 83.5/48 | 25.7/8 |
| STAR | FAIL | 89.2/11 | FAIL | 44.1/10$^*$ |

problems. This motivates a rethinking the goal of neural preconditioner design and the proper model problem for evaluation.

## C   GEOMETRIC MULTIGRID DETAILS

The geometric multigrid (GMG) method is a computational technique used to approximately solve a linear system $Ax = b$ defined on regular grids. We provide more details of our GMG baseline. As illustrated in Alg. 1 and Fig. 4, the V-cycle function is recursively defined over hierarchical grid structures. This process involves three essential components: the smoothing function $S$, the restriction operator $R$, and the prolongation operator $P$. For the smoother, we use the Red-black Gauss-Seidel method (Briggs et al., 2000), which efficiently reduces errors at each grid level. The restriction operator $R$ performs down-sampling, implemented as an average pooling layer that maps, for example, $x^{(l)} \in \mathbb{R}^{256^3}$ to $x^{(l+1)} \in \mathbb{R}^{128^3}$ via standard 8-to-1 average pooling (Shao et al., 2022). The prolongation operator $P$ serves as the transpose of $R$, and performs interpolation-based up-sampling to transfer coarse-grid corrections back to the finer grid.

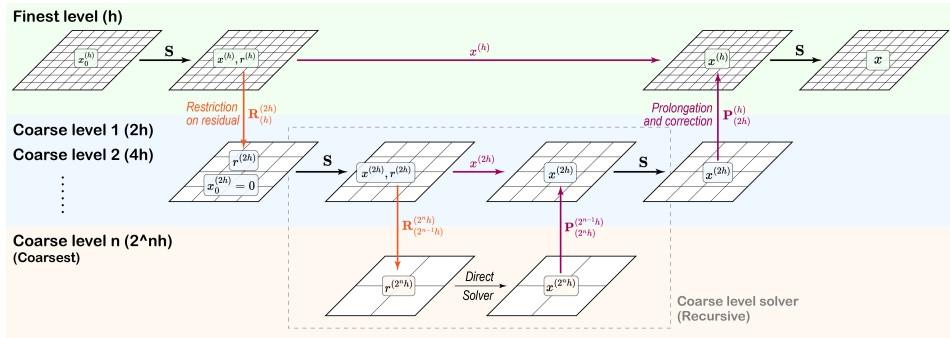

Figure 4: The Multigrid V-cycle scheme.

### C.1   RED-BLACK GAUSS-SEIDEL SMOOTHING

There are various choices of smoothers used in MGPCG V-cycles, such as weighted Jacobi, Gauss-Seidel, SSOR, and Chebyshev smoothers (Tatebe, 1993). In our implementation, we adopt the red-

---

**Algorithm 1** Multigrid V-cycle

---

**Require:** $\boldsymbol{A}^{(l)}$: system matrix, $\boldsymbol{b}^{(l)}$: right-hand side, $\boldsymbol{x}^{(l)}$: initial guess, $l$: current level
**Ensure:** Approximated solution $\boldsymbol{x}^{(l)}$ to $\boldsymbol{A}^{(l)}\boldsymbol{x}^{(l)} = \boldsymbol{b}^{(l)}$

1: **function** V-CYCLE($\boldsymbol{A}^{(l)}, \boldsymbol{b}^{(l)}, \boldsymbol{x}^{(l)}, l$)
2:     **if** $l$ is the coarsest level **then**
3:         **Solve directly:** $\boldsymbol{x}^{(l)} \leftarrow \boldsymbol{A}^{(l)\,-1}\boldsymbol{b}^{(l)}$
4:     **else**
5:         **Pre-smooth:** Repeat $k$ times: $\boldsymbol{x}^{(l)} \leftarrow \text{Smooth}(\boldsymbol{A}^{(l)}, \boldsymbol{b}^{(l)}, \boldsymbol{x}^{(l)})$
6:         **Restrict the Residual:** $\boldsymbol{b}^{(l+1)} \leftarrow \boldsymbol{R}^{(l)}(\boldsymbol{b}^{(l)} - \boldsymbol{A}^{(l)}\boldsymbol{x}^{(l)})$
7:         **Recursive call:** $\boldsymbol{x}^{(l+1)} \leftarrow$ V-cycle($\boldsymbol{A}^{(l+1)}, \boldsymbol{b}^{(l+1)}, \boldsymbol{0}, l+1$)
8:         **Prolongate the Correction:** $\boldsymbol{x}^{(l)} \leftarrow \boldsymbol{x}^{(l)} + \boldsymbol{P}^{(l)}\boldsymbol{x}^{(l+1)}$
9:         **Post-smooth:** Repeat $k$ times: $\boldsymbol{x}^{(l)} \leftarrow \text{Smooth}(\boldsymbol{A}^{(l)}, \boldsymbol{b}^{(l)}, \boldsymbol{x}^{(l)})$
10:     **end if**
11:     **Output:** $\boldsymbol{x}^{(l)}$
12: **end function**

---

black Gauss-Seidel smoother. To provide context, the standard Jacobi iterative method approximates the solution to the linear system $\boldsymbol{Ax} = \boldsymbol{b}$ by repeatedly applying the update $\boldsymbol{x} \leftarrow \boldsymbol{D}^{-1}(\boldsymbol{b} - \boldsymbol{Ux})$, where $\boldsymbol{D}$ and $\boldsymbol{U}$ denote the diagonal and off-diagonal components of $\boldsymbol{A}$, respectively. The red-black Gauss-Seidel method can be viewed as a 2-step block Jacobi iteration, where only half of the degrees of freedom (DoFs) are updated at each step in an alternating fashion.

More concretely, the computational domain is partitioned into red and black cells based on the parity of the sum of grid indices: cells with an odd (even) index sum are labeled red (black). This bipartition works naturally in the finite difference discretizations of first-order and second-order gradient operators, where the resulting matrix $\boldsymbol{A}$ includes off-diagonal only between neighboring grid points in different colors. This 2-coloring induces a bipartite adjacency graph, which resolves the data dependencies inherent in the Gauss-Seidel method and enables efficient parallel in-place updates (Briggs et al., 2000).

During the Red phase (R-phase), black cells remain fixed while only red cells are updated; this is reversed during the Black phase (B-phase). A full smoothing iteration thus consists of the following two updates:

$$\text{Update red cells (R-Phase):} \quad \boldsymbol{x}_R \leftarrow \boldsymbol{D}_{RR}^{-1}(\boldsymbol{b}_R - \boldsymbol{U}_{RB}\boldsymbol{x}_B) \tag{3}$$

$$\text{Update black cells (B-Phase):} \quad \boldsymbol{x}_B \leftarrow \boldsymbol{D}_{BB}^{-1}(\boldsymbol{b}_B - \boldsymbol{U}_{BR}\boldsymbol{x}_R) \tag{4}$$

Here, the subscripts "$R$" and "$B$" refer to the sets of indices corresponding to red and black cells, respectively.

## C.2   BOUNDARY CONDITION AND MATRIX COARSENING

For the heat and Helmholtz equations, we consider problem domains with irregular boundaries and enforce Dirichlet boundary conditions following standard numerical practices (Solomon, 2015). These boundary conditions are typically incorporated by modifying the discretization near the domain boundaries.

In the specific case of the Poisson equation arising in fluid animation, the treatment is more specialized due to the time-dependent evolution of fluid and solid regions. We adopt the boundary handling approach described in prior works (Bridson & Müller-Fischer, 2007; Kaneda et al., 2023; Lan et al., 2024a). Concretely, we classify each voxel grid cell as either a degree-of-freedom (DoF) cell or a non-DoF cell. The unknowns are defined at DoF cells, which correspond to fluid regions, while air and solid regions are marked as non-DoF. During the discretization of the Poisson operator, the local stencil entries of the system matrix $\boldsymbol{A}$ are adapted at the fluid boundaries to appropriately enforce the boundary conditions. See Fig. 6.

**Galerkin Principle**   The multigrid method further requires constructing the coarsened matrix $\boldsymbol{A}^{(l)}$ at each coarser level, which makes handling boundary conditions non-trivial. Several works

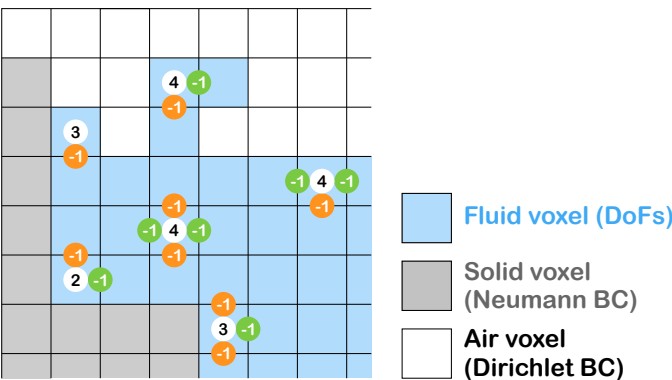

Figure 5: Illustration of boundary condition handling in the discretized operator of the pressure Poisson problems. At boundary cells, the local stencils of the operator are modified to enforce appropriate conditions. This treatment follows the convention used in Bridson & Müller-Fischer (2007); Kaneda et al. (2023); Lan et al. (2024a). Diagonal entries of the system matrix are associated with cell centers, while off-diagonal entries are placed on cell edges. The figure shows the 2D case, which generalizes naturally to 3D.

have discussed strategies for addressing boundary conditions on coarser grids (Briggs et al., 2000; McAdams et al., 2010). We adopt the Galerkin principle to construct the coarsened matrices, which effectively applies a weighted average pooling to the matrix $A^{(l)}$ implicitly stored on a staggered grid. Specifically, we store the diagonal entries at the centers of DoF cells and the off-diagonal entries at the midpoints between adjacent DoF cells. The key features of the coarsening strategy are: (i) it adheres to the Galerkin principle, such that $A^{(l+1)} = P^{(l)}A^{(l)}R^{(l)}$; (ii) when the operator is homogeneous, as in the interior of fluid domains governed by Poisson equations, the coarsened stencil remains consistent with that on the fine grid; and (iii) the coarsening can be performed at negligible computational cost. As a result, irregular boundaries are implicitly handled through this coarsening strategy.

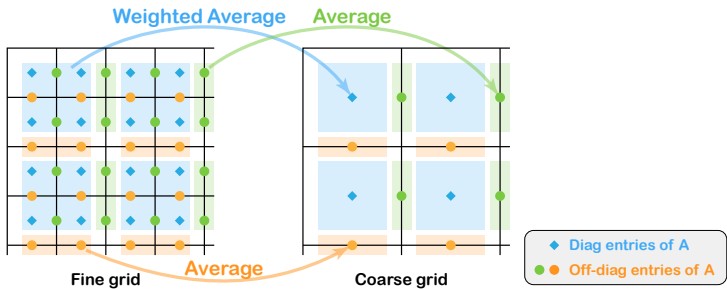

Figure 6: Illustration of matrix coarsening via weighted average pooling.

The above GMG method is the matrix-free, grid-oriented version of the unsmoothed aggregation algebraic multigrid (UAAMG) method (Shao et al., 2022), which is regarded as one of the state-of-the-art (SOTA) GMG solvers for large-scale second-order linear PDE problems defined on regular grids (Shao et al., 2022). We have also tried other GMG variants, including the standard V-cycle GMG method as described in Briggs et al. (2000). However, these methods exhibited stagnation issues when applied to domains with flexible or moving boundaries, a well-known limitation of traditional GMG approaches, as also reported by McAdams et al. (2010).

**Remark** Shao et al. (2022) have discussed and compared 2 CPU-based implementations of the above UAAMG method, one sparse matrix-based version and a matrix-free version. The matrix-based version, originally discussed by Braess (1995), can be generalized to irregular mesh structures and therefore considered as an *algebraic* multigrid method as indicated by its name. Their

matrix-free version is faster but only handles problems defined on regular voxel grids. Therefore, it should be classified as a *geometric* multigrid method, as it does not involve sparse matrix-based implementations or a long precomputation time on building the coarsened matrices.

## D  MODEL DETAILS

Our model is modified based on the dual-channel split (Tatebe, 1993) of the V-cycle scheme (Alg. 2, Fig. 7).

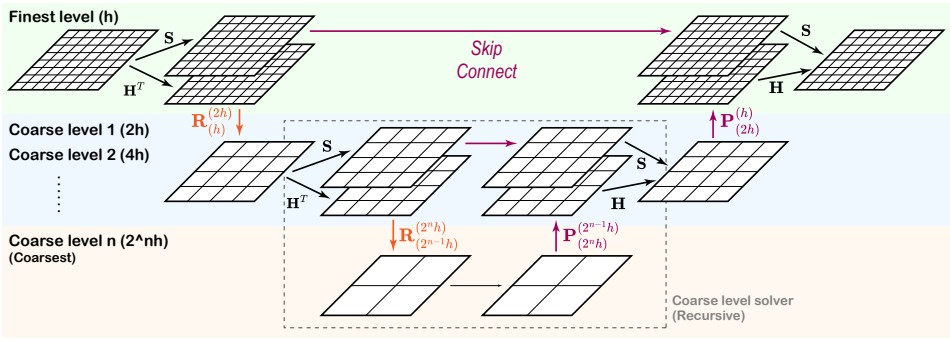

Figure 7: The overview of our model. We follow the dual-channel V-cycle model, but replace the smoothing operator in one channel by a precomputed neural smoother.

---

**Algorithm 2** Our Preconditioner

---

**Require:** $A^{(l)}$: system matrix, $b^{(l)}$: right-hand side, $x^{(l)}$: initial guess, $H^{(l)}$: precomputed convoluional kernels, $l$: current level
**Ensure:** Approximated solution $x^{(l)}$ to $A^{(l)}x^{(l)} = b^{(l)}$
 1: **function** OUR($A^{(l)}, b^{(l)}, x^{(l)}, l$)
 2:      **Classical Channel:**
 3:      **Pre-smooth:** Repeat $k = 2$ times: $x^{(l)} \leftarrow \text{Smooth}(A^{(l)}, b^{(l)}, x^{(l)})$
 4:      **Post-smooth:** Repeat $k = 2$ times: $x^{(l)} \leftarrow \text{Smooth}(A^{(l)}, b^{(l)}, x^{(l)})$
 5:      **Neural Channel:**
 6:      **Apply transposed kernel:** $b^{(l+1)} \leftarrow R^{(l)} H^{(l)\top} b^{(l)}$
 7:      **Recursive call:** $y^{(l+1)} \leftarrow \text{Our}(A^{(l+1)}, b^{(l+1)}, \mathbf{0}, l+1)$
 8:      **Apply learned kernel:** $y^{(l)} \leftarrow H^{(l)} P^{(l)} y^{(l+1)}$
 9:      **Output:** $x^{(l)} + y^{(l)}$
10: **end function**

---

### D.1  RBGS-BASED MATRIX-FREE NEURAL SMOOTHER

We avoid explicitly constructing the neural smoothing operator $H$ for reducing the computational overhead of our neural components. We notice that in the Gauss-Seidel smoothing, the operator $H$ can be decomposed by a series of multiplication of $D^{-1}$ and $U$ 3. Therefore, instead of parametrize the operator $H$, we choose to parameterize the operator $U$ and $D^{-1}$ to composite our neural operator, as deducted by Eq. 3. Rigorously, we compute our parameterized $\tilde{U}(\theta)$ by a 2-layer perceptron, and $\tilde{D}^{-1}(\theta) = (D + \theta_1 I)^{-1} + \theta_2 I$.

We introduce our 2-layer perceptron module Fig. 8 in details. For simplicity, we specify our following discussion on 2D problems, but it is easily generalizable to 3D. We use $\mathbf{I} \in \mathbb{N}^2$ to enumerate the cells, where the unknowns of the linear system are defined. When sub-scripting the matrix $U$, we use $U_{\mathbf{I}}$ to denote the 4 off-diagonal entries in the row of corresponding to cell $\mathbf{I} \in \mathbb{N}^2$. For example, the nonzero entry of its right neighbor is $U_{\mathbf{I},\mathbf{I}+[\mathbf{1},\mathbf{0}]}$, which is at the row of the cell $\mathbf{I}$ and the column of the cell $\mathbf{I} + [\mathbf{1}, \mathbf{0}]$. A kernel of the $\tilde{U}(\theta)$ located at the cell indexed at $\mathbf{I} \in \mathbb{N}^2$ is $\tilde{U}_{\mathbf{I}} \in \mathbb{R}^4$, with

only 4 nonzeros: $\tilde{U}_{\mathbf{I},\mathbf{I}+\boldsymbol{\Delta}}$ for all $\boldsymbol{\Delta} \in \mathbb{N}^2$ such that $|\boldsymbol{\Delta}|_1 = 1$. It is computed as

$$\tilde{U}_{\mathbf{I}}(\boldsymbol{U}_{\mathbf{I}}; \boldsymbol{w}, \Theta) = \boldsymbol{U}_{\mathbf{I}} + \sum_{i=1}^{h} w_i \sigma(\boldsymbol{T}(\boldsymbol{U}_{\mathbf{I}}; \Theta^{(i)})) + w_0 \tag{5}$$

where the operation $\boldsymbol{T}$ is a linear operator that satisfies the equivariance with respect to the axis permutation, designed as the dot produce of the weights and the entries, i.e.

$$\boldsymbol{T}(\boldsymbol{U}_{\mathbf{I}}; \Theta^{(i)})_{\mathbf{I}+\boldsymbol{\Delta}} = [\boldsymbol{U}_{\mathbf{I},\mathbf{I}+\boldsymbol{\Delta}}, \boldsymbol{U}_{\mathbf{I},\mathbf{I}-\boldsymbol{\Delta}}, \sum_{|\Delta'|=1} \boldsymbol{U}_{\mathbf{I},\boldsymbol{I}+\Delta'}, 1] \cdot \Theta^{(i)} \tag{6}$$

and $\sigma$ is an element-wise activation function that introduces the non-linearity.

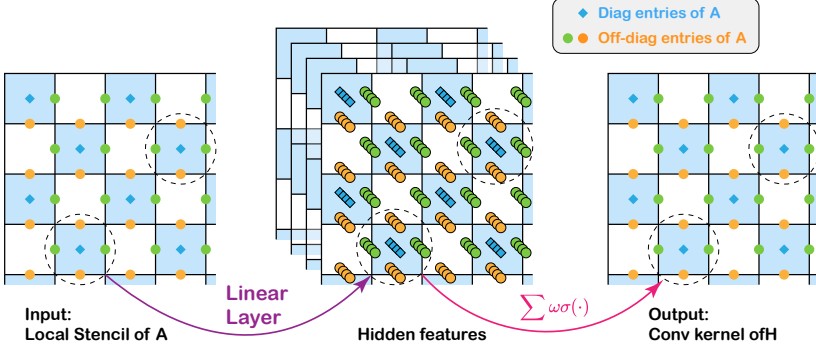

Figure 8: The precomputation of our neural smoothing operator. Note that in red-black Gauss-Siedel, only half (dark-colored) of the cells are activated. Therefore, the mapping is only applied on red (black) cells for the red (black) phase operator.

## D.2 Implementation Details

**Half-top Smoothing** Braess (1995) recommends using only a single smoothing iteration at the finest grid level, rather than two. In practice, we observe that this strategy performs well for Poisson problems: while it may slightly increase the number of Conjugate Gradient (CG) iterations required for convergence, it substantially reduces the total preconditioning time. Accordingly, we apply the half-top smoothing configuration to both our GMG baseline and our proposed model for Poisson problems, ensuring a fair and optimal comparison. For heat and Helmholtz equations, this strategy is disabled due to its limited effectiveness.

**Global Scalar and Over-correction** It is suggested that we should multiply the coarsened solution by a constant scalar $\alpha = 2$ in the prolongation step (Braess, 1995; Shao et al., 2022). We modifies the global constant scalar applied on each level of MG, which, absorbs the following constants: we have omitted a constant scalar $1/8$ when stating $\boldsymbol{R} = \boldsymbol{P}^\top$ (Appendix C), and also in the pooling-based matrix coarsening precomputation, we have omitted a scalar 2 when ensuring the coarsened stencil to be unchanged in the interior DoFs (Sec. C.2). Further, we decide to make the factor learnable in each level and, consequently, we choose to drop global scalar factors in the design our neural layers Sec. D.1.

**Heterogeneous Convolution** For PDEs with uniform stencils in the interior, such as the pressure Poisson problem, our model results in a shared uniform convolutional kernel. Therefore, to reduce the computational overhead introduced by the neural components, at the $128^3$-resolution level, our spatially varying neural mapping is applied only to the cells near boundaries, while a uniform convolution is applied in the fluid interiors. This strategy is not applied to spatially varying PDEs.

**Mixed-Precision** To solve the problem to the residual of 1e-6, we have found that it is insufficient to use 32-bit floating numbers, while it will be unnecessary to use 64-bit floating numbers in the whole computation. Therefore, when applying our hybrid neural preconditioner, all computation

is done in 32-bit floating numbers, while we use the 64-bit floating numbers in the CG iterations outside the preconditioning step. Both our GMG baseline and our model adopt this implementation. This is a setting aligned with Lan et al. (2024a).

# E    EXPERIMENTAL SETUP DETAILS

## E.1    PRESSURE POISSON PROBLEM

For each scene, we select 10 problem instances as training set and the rest for testing. The training set includes 50 instances from 5 scenes in total, and the training typically requires about 2 days (48 hours) to converge using 50,000 CMA-ES function evaluation (traversals of all training data). We test the methods by running our simulator on the fly on a work station with an NVIDIA 4080 GPU.

## E.2    HEAT EQUATION

We have trained two models on the synthetic dataset and our bunny scene separately. The training including 10 problem instances typically requires 10 hours involving 50,000 CMA-ES function evaluation (traversals of all training data) on a server with one NVIDIA A100 GPU. For testing, we test the problem instances including the 300 synthetic ones on the server, and the bunny scene by the work station on the fly with an NVIDIA 4080 GPU.

The heat equation is a time-dependent PDE, and we follow the same standard time evolution strategy as the heat equation testcases in the previous neural PCG work Li et al. (2023). After time discretization, the linear system to be solved in each time step is with the form in Eq. 2, where spatially varying kernels reflect the diffusion coefficients. For the bunny scene, the diffusion coefficients are set to be constant inside the domain, but with smaller value near the bunny surface, which are chosen to mimic a practical scene with different materials for the domain internals and the surface. An illustration of the 2D version of the descritized kernel is $(1 + 4\alpha_{ij})\mathbf{x}_{ij} - \sum_{l=\pm 1} \sum_{k=\pm 1} \alpha_{ij}\mathbf{x}_{i+l,j+k} = \mathbf{f}_{ij}$ where $\alpha_{ij} = \frac{c_{ij}\Delta t}{h^2}$ contains the diffusion coefficients (in practice we use the 3D version). In the synthetic scenes, the diffusion coefficients are randomly chosen and with a fixed spherical obstacles as shown in Fig. 3. The discretized linear system is constructed following by the standard practice in the Poisson case as illustrated in Fig. 6, but additionally with the diffusion value added to the diagonal entries.

In particular, the way of constructing kernels and the choosing of coefficients can be found in our released code, where $c_1 = 5$ and $c_2 = 0.5$ are the domain and the surface choice respectively. We have not included the randomly spatially varying coefficient cases here, as they are separately demonstrated in Appendix F.

## E.3    HELMHOLTZ EQUATION

We have trained on 10 Helmholtz equations with the same coefficient distribution but with randomly generated boundary shapes, and we have tested on 300 instances with different boundary shapes. The training and testing are on the server with one NVIDIA A100 GPU, which requires about 2 hours to converge involving 10,000 CMA-ES function evaluation (traversals of all training data). The tests are done on the server with one NVIDIA A100 GPU.

Helmholtz equations are usually indefinite and not suitable for conjugate gradient-based methods discussed in this work. Therefore, during the construction, we choose a small subset by deliberately choosing a spatially varying wave number that makes the matrix diagonally dominant. Practically, we choose to flip the signs of the Poisson operator, with a large wave number term added on the diagonal entries, chosen to be spatially varying as $c = A\|\mathbf{x} - \mathbf{d}\|^2 + C$. Here $\mathbf{x}$ is the voxel location, and $d$ locates the moving spherical obstacle. A constant $C$ added globally to the diagonal is designed to ensure the matrix diagonal dominant. An illustration of the 2D version of the descritized kernel is $(c - 4)\mathbf{x}_{ij} + \sum_{l=\pm 1} \sum_{k=\pm 1} \mathbf{x}_{i+l,j+k} = \mathbf{f}_{ij}$ where $c > 8$ are chosen to make the system diagonal dominant (in practice we use the 3D version). Therefore, this specific subset of Helmholtz equations are SPD and suitable for conjugate gradient-based solvers. We have not included the randomly spatially varying coefficient cases here, as they are separately demonstrated in Appendix F.

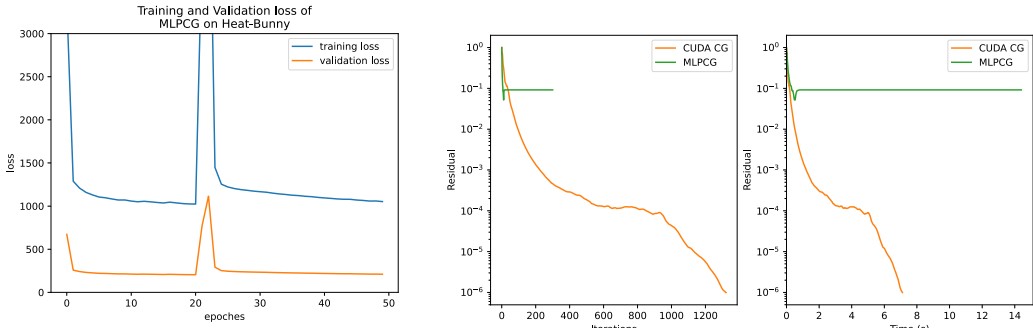

Figure 9: Left: The curve of training and validation loss of MLPCG on a heat equation, from which we can see that the training process has roughly reached convergence with 50 epochs. Right: The residual-iteration and residual-time curves of MLPCG evaluated on exactly the training frame (green), compared with the performance of ordinary CG (orange). The trained model with best loss (at the 20th epoch) did not converge well even on a previously-seen problem.

### E.4  BASELINE METHODS

We use the default settings of AMGCL and AMGX in Lan et al. (2024b) as the compared baseline. We have additionally searched through all 62 configurations in Naumov et al. (2015) to make sure that the best one is compared. We include the precomputation time for AMG methods in the timing, but exclude the time of matrix assembly. Since GMG and our method are matrix-free, there is an extra time in the sparse matrix assembly for AMG methods. We choose to exclude it for a fair comparison, following the same convention as in the previous works (Han et al., 2024; Lan et al., 2024a).

We also choose MLPCG (Lan et al., 2024a) as the baseline for neural preconditioning methods, which is a frame of learning preconditioners for Poisson problems on 2D/3D regular grid domains. We followed their routine of generating 1600 Ritz vectors and sampling 800 RHS from random linear combinations of Ritz vectors. This is a time-consuming task which takes about 1.5h on each frame of our $256^3$ dataset. We then trained the MLPCG model with one single frame from our different datasets.

The performances of MLPCG trained on Poisson equations have been already reported in our paper, and here we give a representative result of MLPCG on heat equations in Fig. 9.

We spent 2 days running 50 epochs of the training process, only to find that the best model failed to solve even the exact problem in the training set. Due to the failure of MLPCG on these variant problems, we only considered solving the Poisson equation with this method in our paper.

To ensure a fair comparison to the MLPCG baseline, we first retrain the model on its dataset and have confirmed its reproducibility. However, the training on our data (with sufficiently many Ritz vectors as suggested by Lan et al. (2024a)) spends an excessive amount of time before observing a convergence CG behavior. Therefore, we instead test its released model on our dataset.

## F  EXPERIMENT ON MORE PDE PROBLEMS

We have added the following spatially varying Poisson problem that further highlights the adaptivity of our model.

We construct the following dataset, solving

$$\nabla \cdot (\alpha(\boldsymbol{x}) \nabla \boldsymbol{u}) = \boldsymbol{f} \tag{7}$$

where $\boldsymbol{u}$ defined on $\Omega = [0,1]^3$ is to be solved, with given source terms $\boldsymbol{f}$ and spatially varying coefficients $\alpha$. We set $\alpha(\boldsymbol{x}) = \sum_{i=1}^{4} m_i \sin^2(w_i \|\boldsymbol{x} - \boldsymbol{y}_i\|) e^{-\|\boldsymbol{x} - \boldsymbol{y}_i\|}$, where the amplitude, frequency, and source locations are uniformly randomly chosen as $m_i \sim \mathcal{U}(0,1), w_i \sim \mathcal{U}(0,100), \boldsymbol{y}_i \sim \mathcal{U}([0,1]^3)$, and the similar setting is applied to $\boldsymbol{f}$. The scene is discretized on the regular grid in

Table 5: The test on the 16M-sized spatially varying Poisson problems on the NVIDIA A100 server. The result is reported in average time (ms) / iterations. The convergence error threshold is set to be 1e-6.

| Setting | GMG | AMGX | MLPCG | UGrid | Ours | Speedup |
|---|---|---|---|---|---|---|
| 10 Training Instances | 352/26.1 | 521/31.0 | N/A | FAIL | **335/24.0** | 1.05× |
| 100 Test Instances | 351/26.2 | 516/31.0 | N/A | FAIL | **337/24.1** | 1.04× |

resolution $256^3$, containing 16 million unknowns to be solved. We have compared our method to numerical baselines and the neural baselines. For the UGrid training, we have generated sufficiently many right-hand-side vectors for each problem instances as suggested (Han et al., 2024). However, we cannot observe a convergence behavior, due to its incapability to handle spatially varying problems. The MLPCG baseline is not applicable, since it is specially designed for fluid pressure Poisson problems only, as it requires an input of types of cells. The boundary conditions are handled the same as Sec. C.2, where we set left and right boundaries to be Dirichlet boundary conditions while the rest Neumann boundary conditions. The result shows that our method consistantly outperforms the baseline methods (Tbl. 5).

## G  EXPERIMENTS ON MORE NEURAL BASELINES

There are also several neural multigrid architectures in recent works, and we'll explain why they're not applicable for our problems and thus not taken into consideration.

### G.1  UGRID

UGrid (Han et al., 2024) is a learning-based multigrid solver that adds intermediate convolution layers in ordinary multigrid methods, in hopes of achieving better performance in eliminating low-frequency errors. We report the performance of UGrid that trained with our Heat, Helmholtz, and spatial-variant Poisson dataset, respectively. A set of 10 presentative frames in the whole simulation is selected as the training set, and we keep all training settings identical to that of the original work. The loss curves of three training processes are plotted in Fig. 10.

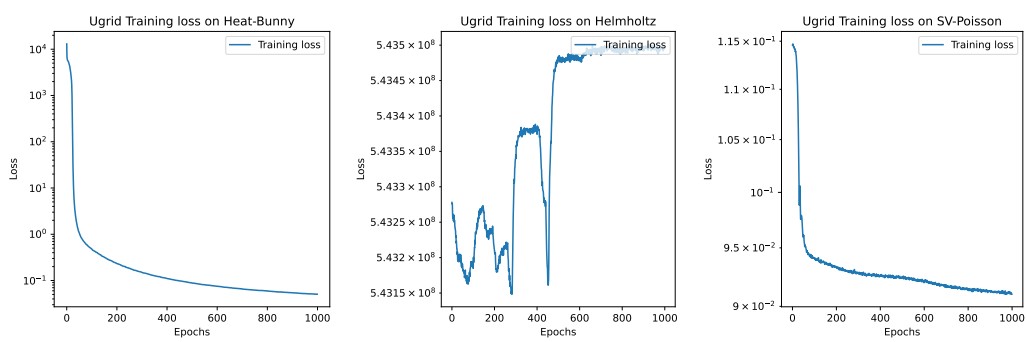

Figure 10: The training loss curve of heat-bunny, Helmholtz, and spatial-variant Poisson equations. The training on the Helmholtz problem fails to converge, while the training on heat-bunny and spatial-variant Poisson leads to a dramatic and non-significant loss decrease, respectively.

We then evaluated the models with frames from the same simulation process, including those in or not in the training set. From the results in Fig. 11, although we can see a significant decrease of the training loss, there is no more performance improvement in solving a real linear system since the 50th epoch. In addition, the final convergence residual of this method is only around $10^{-2}$, which is far beyond our expectations. Cases are worse in the Helmholtz and spatial-variant Poisson datasets, where the training did not converge well.

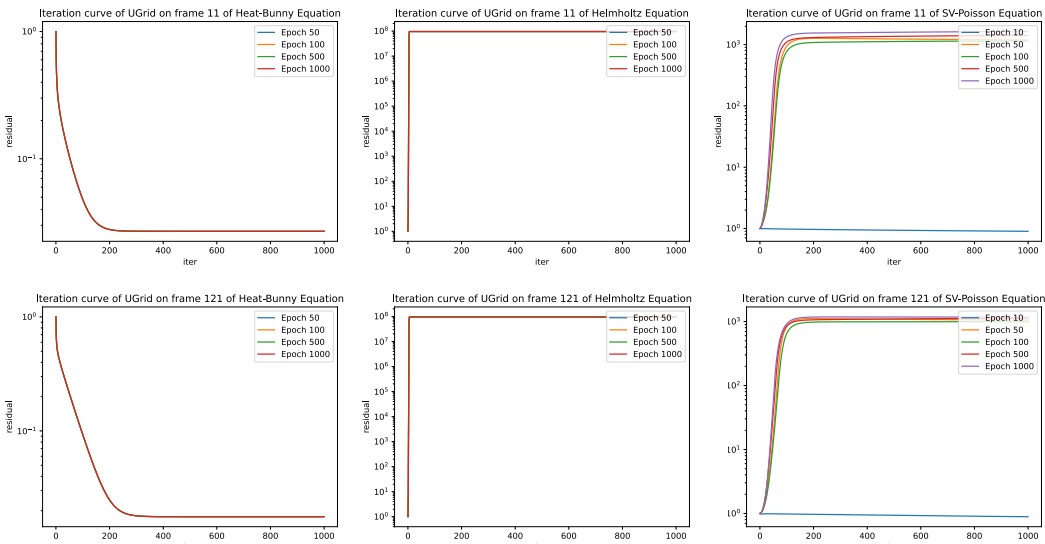

Figure 11: The residual curves of evaluating UGrid models after different epochs of training, tested on seen frames (the first row) and unseen frames (the second row). All colorful curves overlaps in sub-figures 1,2,4,5. The final convergent residual of heat-bunny frames is around $2 \times 10^{-2}$. For the Helmholtz and spatial-variant Poisson scenes, only one initial model (at 10th epoch) tends to preserve the initial residual, while all other models trained on these two scenes fail to converge.

So we may conclude that the UGrid method could not handle these non-standard Poisson equations well, either fails to converge or converges to a solution with low accuracy.

## G.2    MG Smoother Learning

A representative neural multigrid method (Huang et al., 2021) proposes learning smoothers within a pre-defined multigrid hierarchy on uniform grids. While the original work focuses solely on 2D problems, we adapted their up-sampling and down-sampling strategies to 3D domains. However, their coarsening scheme retains 50% of the nodes from $A^{(h)}$ in the coarsened level $A^{(2h)}$, resulting in a substantial computational and memory burden. This becomes particularly prohibitive for large-scale problems, such as those with grid resolution $256^3$. In our case, their construction of coarsened matrices exceeds the memory capacity of a single NVIDIA A100 GPU. Given these constraints, this method is not suitable for our large-scale datasets and was therefore excluded from our experimental comparisons.

## G.3    Neural Helmholtz Solver

Additionally, there have been recent works focusing on large-scale challenging Helmholtz equations (Cui et al., 2025), reporting an 300-second (300,000 milliseconds) solving time on 16M-DoF problems. However, we have found that our method and theirs are not directly comparable. They specifically focus on a class of Helmholtz equations with high wave numbers and fine-grained spatial discretization, making the whole matrix non-SPD and extremely challenging. Our method is not designed for those PDEs, since the PCG-based method fails on non-SPD problems. On the other hand, their method cannot be trivially applied to our testcases, since their model explicitly takes the wave numbers as inputs.

Table 6: The generalizability of our methods to new simulation scenes on Poisson problems. We include the data of 5 scenes from our fluid Poisson dataset as the training set, and test on a new scene "Worm". The "Dambreak-2M" is a dambreak simulation in $256^3$ resolution with about 2 million unknowns in each equation, while "Dambreak-16M" and "Dambreak-64M" are in $512^3$ resolution with 16 and 64 million unknowns for each equation respectively.

| Training Scenes | Test Scene | GMG | AMGX | Ours | Speedup over GMG |
|---|---|---|---|---|---|
| 5 Poisson (Our) scenes | Worm | 174 | 737 | **159** | $1.09\times$ |
| Dambreak-2M | Dambreak-16M | 263 | 462 | **217** | $1.21\times$ |
| | Dambreak-64M | 601 | 1251 | **476** | $1.26\times$ |

Table 7: Comparison of CG iteration counts and solving time across three unseen deformable trajectories of the fish scene without re-training.

| CG Iter / Solving Time (ms) | Trajectory 1 | Trajectory 2 | Trajectory 3 |
|---|---|---|---|
| Ours | 13.82 / 221.6 | 13.43 / 215.8 | 14.29 / 229.2 |
| GMG | 15.73 / 239.1 | 15.26 / 233.1 | 16.13 / 247.0 |

# H ABLATION STUDY

## H.1 ABLATION STUDY ON THE NEURAL CHANNEL

Our model with initial parameterization is mathematically equivalent to a GMG V-cycle and inherits its convergence rate with small computational overhead with typically 1.7ms in precomputation and 0.4ms per CG iteration. After the training, the average CG iteration is reduced by 1-2 on Poisson equations and by more than $2\times$ equations, contributing to a faster convergence rate and shorter solving time (Tbl. 8).

## H.2 LOSS DESIGN

In previous works, there have been long discussion on how to train a preconditioner automatically. We view that the key property of the loss function is whether the decrease in the loss function reflects a fast CG solving. We mainly tested two categories of loss functions in our work, namely the right-hand-side-oriented and the matrix-oriented method.

The trivial way to compute the L2 loss is that

$$\mathcal{L}_2(\boldsymbol{M}; \boldsymbol{A}, \boldsymbol{b}) = \|\boldsymbol{A}^{-1}\boldsymbol{b} - \boldsymbol{M}\boldsymbol{b}\|_2^2 \tag{8}$$

and also the residual L2 loss giving

$$\mathcal{L}_r(\boldsymbol{M}; \boldsymbol{A}, \boldsymbol{b}) = \|\boldsymbol{A}\boldsymbol{M}\boldsymbol{b} - \boldsymbol{b}\|_2^2 \tag{9}$$

Table 8: Ablation study results on our simulation datasets. "Init." refers to our model before training. Each entry reports the average computation time (ms) and the average CG iteration over the entire simulation segment. Bold values indicate the best performance.

| DATASET | SCENE | GMG | INIT. | OURS |
|---|---|---|---|---|
| FLUID (OURS) | BALL | 172/15.1 | 175/15.1 | **156/13.6** |
| | PROPELLER | 167/15.0 | 173/15.0 | **156/13.5** |
| | FISH | 175/15.7 | 181/15.7 | **159/13.8** |
| | ROBOT | 171/15.4 | 177/15.4 | **157/13.6** |
| | SPIN | 185/16.7 | 191/16.7 | **179/15.6** |
| HEAT | BUNNY | 641/39.7 | 658/39.7 | **135/7.7** |

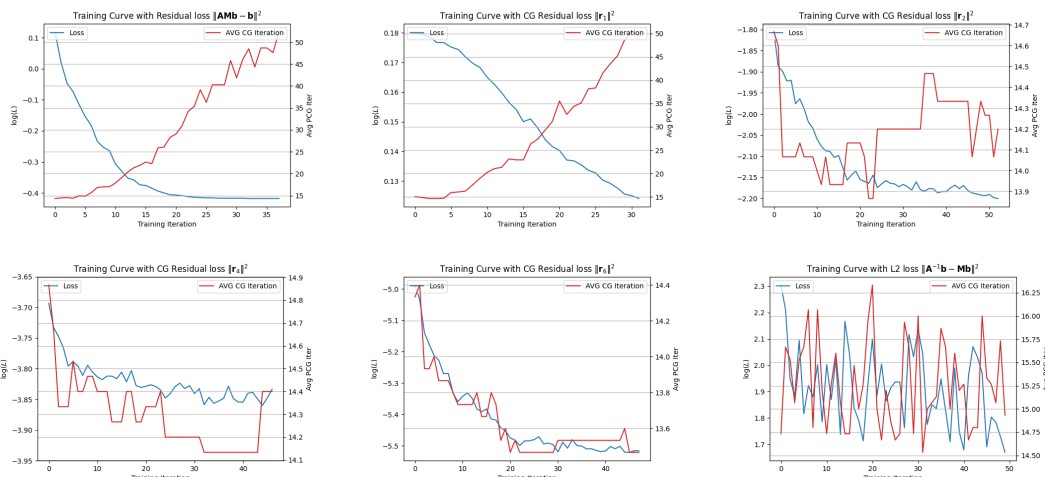

Figure 12: Training and validation curvers under different loss designs.

Table 9: Repeating the "generalization to a new scene" setting to analysis the variance. We train on 5 fluid Poisson scenes and testing on the unseen "Worm" scene. The standard error is reported. We report the variance across different frames in one test of a trained model at the column "Across 300 Instances". The variance of running tests for the same model is reported in the column titled "Across 3 Tests". The rightmost column is the average and standard error of the frame-averaged metrics across 3 different trained models.

| Metric | Across 300 Instances | Across 3 Tests | Across 3 Trainings |
|---|---|---|---|
| Solving Time (ms) | $159.4 \pm 5.2$ | $160.0 \pm 0.46$ | $160.6 \pm 0.88$ |
| CG Iterations | $13.80 \pm 0.43$ | $13.80 \pm 0$ | $13.84 \pm 0.04$ |

Our choice is the residual at the $k$-th CG iteration.

It is shown in Fig. 12. The x-axis refers to the CMA-ES iteration, where each iteration involves multiple function evaluations. The CG iterations are tested in 10 validation frames. Among all of the loss function designs, we find that only the $k$-th CG iteration loss with $k \geq 6$ align with the CG iteration drops best. Other loss designs result in divergence in CG or involves instability in the training.

There are also matrix-oriented loss functions that we did not try for their heavy computational costs. Those loss functions can provide theoretical guarantees to the CG convergence rate. For example, the minimization of the condition number directly bounds the convergence rate of CG in the worst case. However, the matrix-oriented loss designs suffer from the computational difficulties for large-scale problems. An alternative choice proposed is to use the L2 loss on the Ritz vectors of the matrix instead of the dataset of right-hand-side vectors. However, it consumes more computational efforts on factorizing out the Ritz vectors on the large-scale problem in resolution $256^3$.

## H.3 VARIANCE ANALYSIS

We classify the variance of our model performance into three aspects: (i) the variance across different frames in one simulation scene; (ii) the variance when repeatedly running the test script, which introduces the variance of the timing but does not affect the CG iterations; (iii) the variance of training with different random seed. We choose our "generalization to a new scene" setting to analyze the variances.

As shown in Tbl. 9, our model consistently yields performance improvements across different training and testing runs. The variance in average solving time remains below 1 ms, supporting the statistical significance of our minimum $1.05\times$ speedup.

## H.4 CG Performance on the MLPCG Dataset

We further justify our result by directly applying our GMG baseline as well as our method on the test problems in the previous works (Lan et al., 2024a).

We provide the CG performance curves on the key frames in Fig. 13. In all of the CG residual-iteration curves, the AMGCL, our GMG baseline, and our method converges to 1e-6 relative error within 20 CG iterations, which is the aligned with the expected performance of a multigrid preconditioner. However, the AMGCL method requires an extremely long precomputation time, and even fails at the frame 188 of the torus scene. In contrary, both of our method and the GMG baseline takes negligible precomputation time. For the AMGX method, we adopt the default classical Jacobi setting, which is faster than the customized setting compared in Lan et al. (2024a). Compared with our method, they spend more CG iterations and consumes even longer time in each preconditioning.

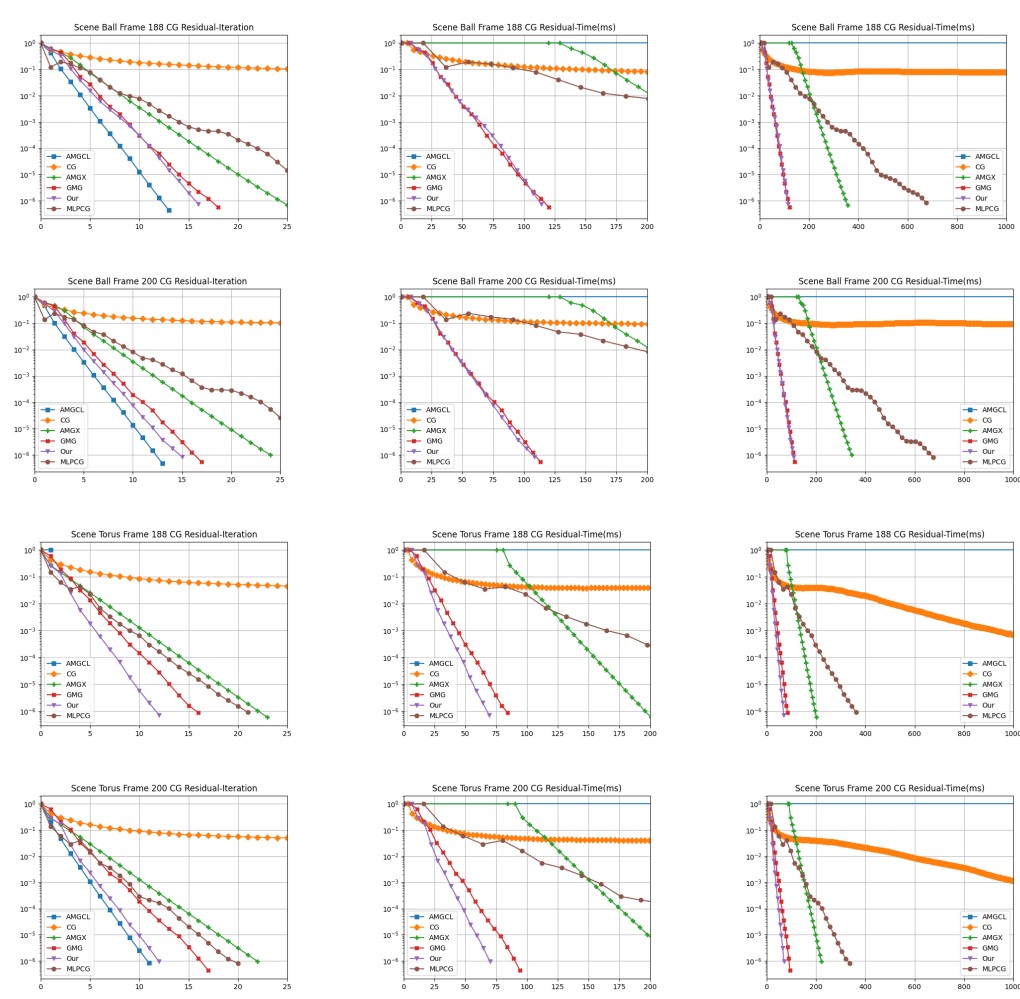

Figure 13: Conjugate gradient residual curves on different frames in the MLPCG dataset (Lan et al., 2024a). "CG" refers to not applying preconditioners. The relative residual versus the iteration and time in millisecond (including the precomputation time but excluding the matrix assembly) is reported. The right column has a larger range in the time axis than the middle column without other changes.

Table 10: Additional generalizability tests on cross-scene / cross-resolution settings. Here 56M scenes are with $384^3$ cells.

| Training Scenes | Test Scene | GMG | Ours |
|---|---|---|---|
| 5 Poisson (Our) scenes | 4K ShapeNet airplanes | 183ms / 16.6 CG Iters | **166**ms / 14.9 CG Iterations |
| Heat Bunny 16M | Heat Bunny 56M | 1406ms / 32.7 CG Iterations | **361**ms / 8.9 CG Iterations |
| SPD Helmholtz 16M | SPD Helmholtz 56M | 1238ms / 28.0 CG Iterations | **941**ms / 21.0 CG Iterations |

## I  GENERALIZABILITY

We have tested the generalizability of our model from two aspects: to a new scene with unseen boundary geometries and to the same scene with a larger resolution referring to Tbl. 6.

We additionally demonstrate the generalizability of our method on all 4,045 voxelized airplane models from ShapeNet Chang et al. (2015), using them as obstacles in the inlet-flow tests. These airplane geometries are entirely unseen during the 5-scene Poisson training. The averaged performance results are summarized in Tbl. 10, where improvements are observed in 3,885 out of 4,045 scenes. Only 17 out of 4,045 cases exhibit performance more than 50 ms slower per instance compared to the GMG baseline, indicating that tailoring to the specific data distribution may sacrifice some corner cases uncovered.

We have also provided the cross-resolution generalizability tests for the heat and Helmholtz equations in Tbl. 10.

## J  VISUALIZATION

The visualization of our dataset is demonstrated in Fig. 14. Each scene in Fig. 14 represents a simulation process, and each involves solving more than 200 different equations. All these scenes, except for the dambreak, are run with a grid resolution of $256^3$. The dambreak scene is simulated with grid resolution of $256^3$ and $512^3$, respectively. The simulation timestep is determined by the CFL condition, and the condition number is 5.

**Fish**   An inlet flow is placed at the boundary of the simulation domain, and a fish-shaped object moves toward the inlet with a swimming-like motion. The vorticity field of the simulation is visualized with red and blue color, each color represents different direction of vorticity.

**Worm**   An inlet flow is located at the boundary. A thin, worm-like strip moves with the inlet flow, waving its body and disturbing the fluid within the simulation domain. The vorticity field is also visualized with red and blue color.

**Robot**   In this scene, an inlet flow is set at the right-most boundary of the domain. A four-leg robot moves with the inlet flow and mimics the octopus's movement. The vorticity field is visualized with a red-to-yellow color field, representing increasing vorticity strength.

**Spin**   In this scene, a spinning cylinder with four pipes is placed at the bottom of the simulation domain. Smoke continuously streams upward under the cylinder, goes through the four pipes, and spreads inside the domain.

**Ball**   In this scene, a smoke source is placed at the leftmost boundary of the simulation domain. The smoke leaks out with initial velocity from left to right, then interacts with an oscillating up and down ball.

**Propeller**   In this scene, smoke is released from 3 different locations. A spinning propeller is placed in the center of the simulation domain and interacts with the leaking smoke.

**Dambreak** In this scene, a square of liquid is initialized inside the simulation domain. When the simulation starts, the liquid moves dynamically within the domain, driven by gravity force and the repeatedly impact from a baffle on the right side.

**Heat-bunny** In this scene, we place a bunny-shaped shell inside the domain, and the shell transfers heat slower than the other area. A moving sphere acts as a heat source inside the shell. Consequently, the temperature inside the shell rises rapidly, while the temperature outside the shell warms up gradually.

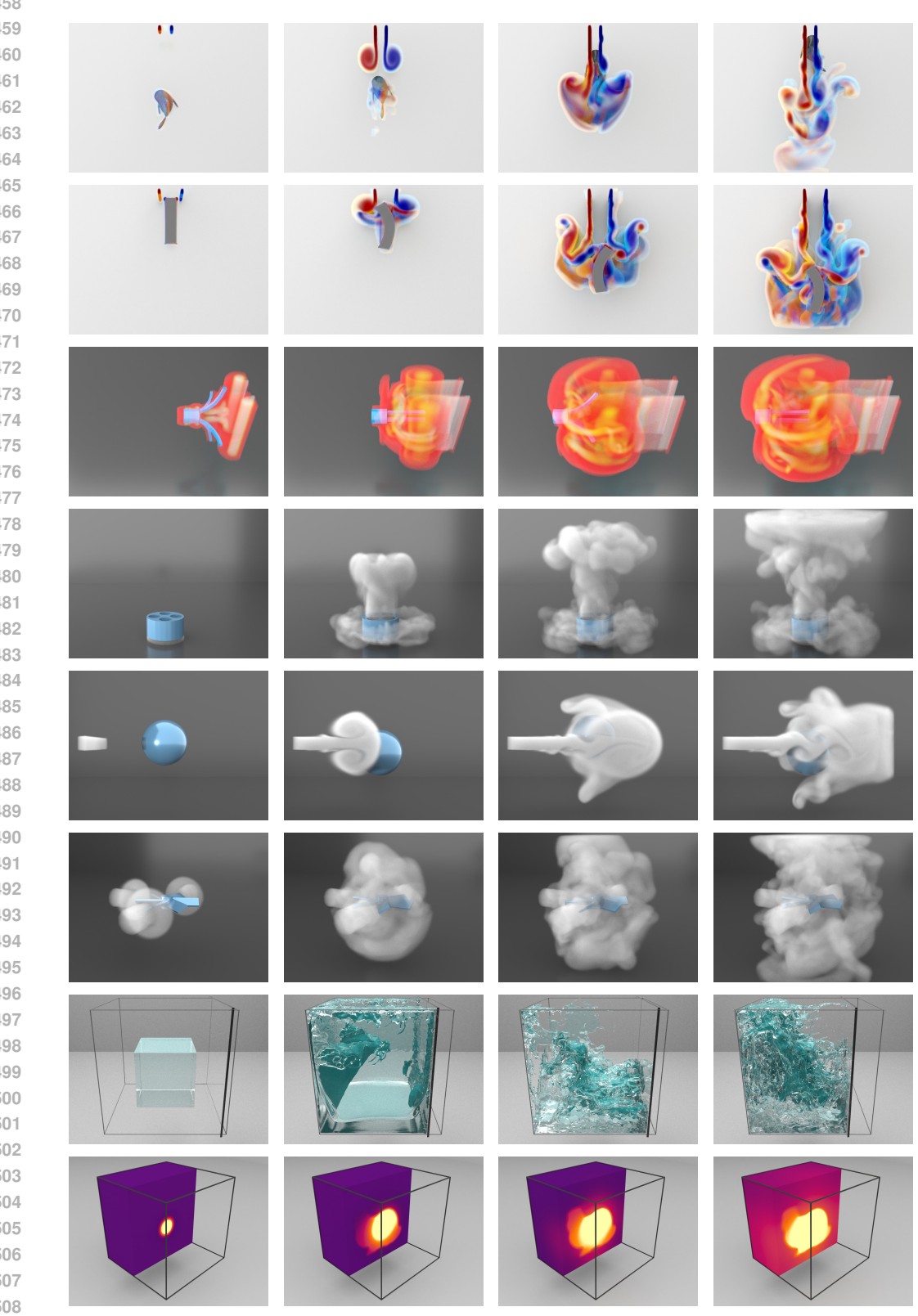

Figure 14: Visualization of the dataset. From top to bottom: fish, worm, robot, spin, ball, propeller, dambreak, and heat-bunny. Detailed description of each scene is in Appendix J.

