# OpenReview forum: "Closing the Performance Gap in Neural Conjugate Gradient Method: A Hybrid Multigrid Preconditioning Approach"
_ICLR.cc/2026/Conference — Submitted to ICLR 2026_

### Official Review · Reviewer_FMaR · 2025-10-29

**Soundness:** 4
**Presentation:** 4
**Contribution:** 4
**Rating:** 6
**Confidence:** 4

**Summary:**

This work presents an efficient neural preconditioner that outperforms existing SOTA neural solvers as well as geometric and algebraic multigrid preconditioned solvers on large scale linear systems arising from different physics problems. This so-called dual-channel neural preconditioner solver preserves symmetry and positive definiteness of conjugate gradient solver. The model is validated on Poisson equations, heat equations and Helmholtz equations.

**Strengths:**

- The results are very impressive, considering geometric multigrid as a strong baseline, and the speedup is demonstrated through extensive experiments.
- This paper addresses the drawback of most neural solvers: unpredictable convergence rate. Since its framework is built upon a traditional multigrid V-cycle, it preserves the SPD of the V-cycle, and so convergence rate is guaranteed.
- It scales very well to large problems in various PDE problems.

**Weaknesses:**

- Since the neural operator is CNN-based, it does not take account of *sparsity* of a linear system. This model likely would not offer much benefit on large *sparse* linear systems.
- This model is limited to first-order grid-cell discretization. It does not address other discretization schemes or non-structured grids.
- This problem probably comes with any solver that involves down- and up-sampling: since coarsening halves size of the grid on each dimension, what would you do for a domain with odd dimension, eg, $511^3$.

**Questions:**

- Did you experiment with different kernel sizes for the convolution operators?
- What is your intuition on choosing *k-iteration* residual as your RHS and loss function to train your network? For one matrix, do you use a fixed k-iteration (hence one RHS vector) or various iterations (more than one RHS)?
- What is your definition of *expressiveness* for neural solvers?
- I need help understand why the neural smoothing convolution operator is *SPD*, especially why it is positive definite?
- What is the stop condition for training?
- Do you have a report of GPU usage when training your models? Does the model of $512^3$ fit on a single GPU?
- Is there any evidence that the proposed model actually improved *condition number* of the linear systems?
- Did you train a separate model for each scene? Does the model generalize to differene scenes?

---

> ### Author Response · Authors · 2025-11-23
>
> We thank the reviewer for the support and constructive comments. Below we provide additional clarifications.
>
> ## Application to Large Sparse Linear System
> Our work, consistent with prior research in this line (FluidNet [ICML 2017], DCDM [ICML 2023], MLPCG [ICML 2024]), focuses on sparse linear systems defined on regular grids rather than dense ones.
>
> ## Clarification on “Expressiveness”
> By *expressiveness*, we refer to the capacity of a parameterization to generate a broad family of valid preconditioners for a given matrix. Limited parameterization constrains this space, while richer and more structured parameterization enlarges it.
> Within this perspective, the proposed neural smoother is more expressive than the AffConv layer in MLPCG because our method exhibits stronger parameterization and nonlinear dependence on $A$, whereas AffConv remains largely linear.
>
> ## Handling Odd Number of Cells
> As demonstrated in our released code, a standard and practical strategy for domains with an odd number of cells is to treat appropriate surrounding cells as boundary cells. The resulting irregular boundaries are coarsened following the procedure detailed in Appendix C.2, and the smoother is trained to adapt to these boundary regions. This maintains a consistent multigrid hierarchy and stable convergence behavior.
>
> ## Large Kernels
> We have tested the use of larger convolution kernels. In 3D, enlarging the kernel substantially increases the computational cost per iteration, and the modest reduction in iteration count does not offset this overhead.
> In particular, extending from 1-hop to 2-hop neighborhoods increases the computational complexity approximately as $O(k^3)$ for kernel side length $k$. Further, larger kernels require multi-coloring instead of two-coloring, totally increasing the per-iteration cost by an order of magnitude. Since vanilla MGPCG typically converges in roughly 10 iterations, such additional cost is unlikely to result in an overall improvement. Based on these facts, the chosen kernel size provides the best balance between efficiency and performance, while efficient large-kernel implementations remain an interesting direction for future exploration.
>
> ## Cross-scene Generalizability
> The revised version includes further training details in Appendix I. As demonstrated in the new ShapeNet experiments, the proposed method performs robustly ($10%$ average speedup on 4K unseen models) under cross-scene evaluation without retraining.
>
> ## Choice of $k$-Iteration
> As explained in Appendix F, the number of iterations $k$ is selected empirically as the minimal value that achieves consistently strong performance across training cases.
>
> ## SPD Guarantee
> The proposed preconditioner $\mathbf{V}$ is linear and symmetric by construction, and it can be derived that $\mathbf{V} = \mathbf{V}^\top$. This intuitionally follows from the use of the transposed kernel during the post-smoothing phase, as detailed in Eqn. (1). The two parts of Eqn. (1) are both symmetric, and is SPD in the initial unparamterized setting.  Although the SPD property, as an inequality constraint, may be lose during training, we refer to the method as *empirically* SPD because training enforces convergence across all training instances, which strongly encourages the resulting preconditioner to be SPD in practice.
>
> ## GPU Usage and Other Details in Training
> In the current gradient-free training setup, GPU memory usage matches that of the inference phase because no backpropagation buffers are required. For a $256^3$ instance, the memory footprint is approximately 1.4 GB. The model at $512^3$ will require 12GB memories.  Additional training details, such as iteration used in training, are included in the revised manuscript. The stopping condition is chosen empirically as the loss seems to converge.
>
> ## Summary
>
> We appreciate the reviewer’s thoughtful feedback and look forward to your continued support. We are happy to address any further questions or concerns.

---

> > ### Comment · Reviewer_FMaR · 2025-11-24
> >
> > Thank you for your clarification. I am overall positive with this work. I will keep the score.

---

### Official Review · Reviewer_gCMy · 2025-10-31

**Soundness:** 3
**Presentation:** 2
**Contribution:** 2
**Rating:** 4
**Confidence:** 4

**Summary:**

Authors consider a problem of learning neural preconditioner for conjugate gradient method. The proposed neural preconditioner is a multigrid method with learnable smoother parametrising with the neural network in such a way that the resulting operator remains a linear iterative method.

The resulting method is successfully applied to Poisson, heat and Helmholtz equations.

**Strengths:**

Authors perform a set of large scale experiments in $D=3$. The number of degrees of freedom significantly exceeds typical experiments on learning and adaptation of multigrid methods making results more convincing and more realistic.

**Weaknesses:**

Many similar results are available in literature, so novelty of the proposed approach (besides the scale of experiment) is questionable. Besides that, there may be issues with diversity of train and test data.

**Questions:**

**Novelty**

Authors describe their research as completely novel: "the first demonstration that a neural preconditioner can surpass highly optimized MG methods on systems with tens of millions of variables", "we have shown that the carefully tuned geometric MG preconditioners
remain strong baselines, but our method surpasses them for the first time", etc. There are dozens of related works that target similar problem--improvement of multigrid components given some family of equations:
1. Multigrid acceleration techniques and applications to the numerical solution of partial differential equations, J Zhang. Application of adaptive multigrid that computes relaxation parameters based on the performance of the method.
2. Automating the Design of Multigrid Methods with Evolutionary Program Synthesis, https://arxiv.org/abs/2312.14875. Design of multigrid algorithms (including preconditioners) with evolutionary algorithms.
3. Learning to Optimize Multigrid PDE Solvers, https://arxiv.org/abs/1902.10248. Direct learning of restriction and prolongation operators of geometric multigrid for parametric PDE problems.
4. **Mentioned by authors.** Learning algebraic multigrid using graph neural networks, https://arxiv.org/abs/2003.05744. Work by the same group but this time restriction and prolongation operators are learned from algebraic multigrid.
5. Neural Multigrid Architectures, https://arxiv.org/abs/2402.05563. Learning restriction, prolongation operators and parameters of smoothers for geometric multigrid.
6. Deep Multigrid: learning prolongation and restriction matrices, https://arxiv.org/abs/1711.03825. Optimisation of restriction and prolongation matrices.
7. **Mentioned by authors.** Learning optimal multigrid smoothers via neural networks, https://arxiv.org/abs/2102.12071.

Authors downplay several contributions mentioned above based on the small scale problems ($\simeq 10^{5}$ variables) considered in these publications. I find this framing unfair: (i) in several publications, e.g., [3], [4], [5] authors train on problems of specific size and confirm that the method performs well for problems of larger size; (ii) evaluation on small size still enough as a proof of concept.

Can the authors explain why they believe their experiments are sufficiently novel and interesting when many very similar contributions are available in the literature?

**Training details**

In Appendix E authors summarise training details. The description here is fragmentary, so I have many questions.

*Details on equations, discretisations, etc*

1. What is the pressure Poisson problem precisely? Can the authors provide details on boundary conditions, discretisation, integrators they use for Navier-Stokes or other equations they used to model fluid flow?
2. Similarly, it is not clear how the heat equation is discretised, what thermal conductivity used, etc. Can the authors provide more details?
3. The same goes for the Helmholtz equation. Please, specify boundary conditions, discretisation, how wave numbers were selected, etc.

*The data on the number of train and test instances is unusually small.*

The number of train and test data is reported in appendix and in the main text

| Equation  | Dataset size                              | Training time |
| --------- | ----------------------------------------- | ------------- |
| Poisson   | 2000 equations, 50 for train              | 2 days        |
| Heat      | 300 equations, 10 for train, 290 for test | 10 hours      |
| Helmholtz | 300 equations, 10 for train, 290 for test | 2 hours       |

1. It is not entirely clear whether authors report the number of equations of the form $Ax = b$, or the number of rollouts of numerical integrators for evolution PDE. Can the authors clarify this? In addition if the later or rollouts is reported, can the authors provide the number of linear systems $Ax = b$ the train on?
2. The dataset sizes are unusually small, and it makes it unclear how well neural networks trained on such problems are going to generalise. I kindly ask the authors to explain why the training data is sufficient and what is a practical significance of selected training setup.
3. Training time is clearly not great, likely owing to the unusual gradient-free optimisation strategy. It is clear that overall the approach is not going to scale well, when more diverse datasets are considered. Can the authors comment on that?

**Helmholtz equation**

Vanilla multigrid is not efficient as a preconditioner for the Helmholtz problem. The search for good preconditioners is still active for the Helmholtz equation with many different options proposed in the last 30 years. A notable strategy is to apply complex shift to the Helmholtz operator:
1. On a class of preconditioners for solving the Helmholtz equation YA Erlangga, C Vuik, CW Oosterlee
2. How large a shift is needed in the shifted Helmholtz preconditioner for its effective inversion by multigrid? PH Cocquet, MJ Gander
3. Preconditioning Helmholtz linear systems, D Osei-Kuffuor, Y Saad
There are also other strategies, e.g., sweeping preconditioner, perfectly matched layers. Did the authors consider preconditioners specialised for the Helmholtz equation? Why did authors decide to compare with multigrid, which is known to perform poorly for this particular problem?

**Smoothers and parallelisation**

Authors claim that GS smoother is highly parallel. This is not the case. In fact, GS smoother is hard to parallelise and because of that polynomial smoothers (Chebyshev) are often preferable, see Parallel multigrid smoothing: polynomial versus Gauss–Seidel, https://www.sciencedirect.com/science/article/abs/pii/S0021999103001943. It may be the case that authors mean a specific read-black Gauss-Seidel smoother, which is actually a Jacobi method applied on a staggered grid.

Polynomial smoothers also potentially form a stronger baseline for geometric multigrid method, see Local Fourier Analysis of Multigrid Methods with Polynomial Smoothers and Aggressive coarsening, https://arxiv.org/abs/1310.8385.

It would be helpful if authors could report the performance of polynomial smoothers (with appropriate prolongation and restriction) for spd matrices in their benchmarks.

**Minor points**

1. Authors reference (Nocedal & Wright, 2006) for CG and (Evans, 2022) for PDE discretisation. I find both these references to be not entirely relevant: (i) for CG it is easy to find original paper where the algorithm was introduced; (ii) the book of Evans is not on numerical solution of PDEs.
2. Algorithm 2 in Appendix D.
   a. Why does post-smoothing directly follow pre-smoothing? In the multigrid scheme it is applied after coarse grid correction.
   b. Why is there no computation of residuals in the whole algorithm? In the current version $x^{(l)}$ is computed from smoother and never used after that.

---

> ### Author Response · Authors · 2025-11-24
>
> We address the reviewer's concerns as follows.
>
> ## Novelty and Neural MG Baselines
> Our work focuses on the recent advancement in neural conjugate gradient methods targeting typically on Poisson problems in fluid animation, a setting established by the previous work FluidNet [ICML2017], DCDM [ICML2023], and MLPCG [ICML2024]. We additionally provide some heat diffusion equations and a subset of SPD Helmholtz equations to investigate the application boundaries of the neural CG methods.
>
> Because most referenced baselines do not provide code, a direct comparison is not available, but our measurements show that even a single sparse matrix–vector multiplication at the problem scale (via cuSparse) is already slower than a full V-cycle of our method, making all methods that predicts sparse matrix-based prolongators noncompetitive in our problem settings. A very conservative approximation of the computational overhead is given  as follows
>
> |                                                     | Inference Time Overhead at 256^3 | Per-iteration overhead over strong GMG at 256^3 |
> | --------------------------------------------------- | -------------------------------- | ----------------------------------------------- |
> | Our                                                 | 2ms                              | ~1ms                                            |
> | RL-AMG [Taghibakhshi2021] | >100s         | >20ms      |
> | Other Prolongator prediction methods | >10 ms                          | >20ms                                           |
>
> There are other methods like program synthesis cited by the reviewer on searching the structure of the MG cycles mentioned by the reviewer. They are orthogonal and complementary to our contribution, as both structure and certain parameters can be jointly optimized.
>
> ## Clarification on Equation Instances
>
> The term "fluid Poisson" refers to solve the pressure projection step in the stable fluid animation (https://www.cs.ubc.ca/~rbridson/docs/batty-siggraph2007-variationalcoupling.pdf), and the way of discretization and  other settings are exactly the same with the previous work DCDM [ICML 2023] and MLPCG [ICML 2024]. For more details, please refer to our revised Appendix E.
>
> ## Clarifying Algorithm 2
>
> We have confirmed that our derivation is correct and have provided more derivations in the revised version. It is an implementation of Eqn. (1) with both branch paramterized, and therefore the $\mathbf{x}$ is used in the output line and no residual computation is required. We are happy to answer more detailed questions if this remains to be your concern.
>
> ## Helmholtz Equation
>
> Indefinite systems induced from the Helmholtz equation is generally hard requiring decades of developments of numerical solvers as mentioned by the reviewer. We would like to clarify that  **our goal is not to outperform specialized numerical Helmholtz solvers on difficult, indefinite systems.** In contrast, our intention is to demonstrate the potential of neural methods and explore the boundaries for neural CG. The SPD Helmholtz case serves as one of the examples within the application boundary of CG-based neural methods, achieved by choosing a sufficiently large wave number $k^2$ in $(-\Delta -k^2)u=f$ so that the discretized system is diagonally dominant by $k^2$. In this particular subset of Helmholtz equations, we empirically find that the MG remains to be the competitive baseline.
>
> ## Chebyshev Polynomial Smoothing
>
> In our manuscript, the Gauss-Seidel method specifically refers to **red–black Gauss–Seidel (RBGS)**, which is well suited for CUDA parallelization and highly memory-efficient due to its in-place update structure.
>
> As noted in work cited by the reviewer, achieving a convergence rate comparable to a 2-iteration RBGS multigrid (MG) smoother requires using a Chebyshev polynomial of at least order 2. However, such 2nd order smoothing is already too costly involving at least 3 SAXPY operations and 3 matrix-vector products in each smoothing, resulting in 6ms at 256^3 scale under our current Taichi implementation, which is the same to the overall V-cycle time consumption.
>
> These observations support our choice of RBGS (as well as [Hu2021], [Shao2022]) as a strong and efficient baseline for comparison.
>
> ## Training Set Size
>
> Our training set setting is similar to the previous work DCDM [ICML 2023] and MLPCG [ICML 2024] targeting mainly on fluid Poisson in this line. In DCDM, authors only train on the square domain, while in MLPCG, authors include 10 fluid scenes ($128^3$) with 10 matrices used in each scene. The training setting, namely 5 scenes with 10 matrices each, is similar as the previous settings. We have also provided additional generalizability test, showing that such training set is already pretty effective.
>
> ## Summary
>
> We hope the above responses address the main concerns regarding the novelty and contributions of our work, and we welcome any further questions the reviewer may have.

---

> > ### Comment · Reviewer_gCMy · 2025-11-27
> > **follow up**
> >
> > I thank the authors for a detailed reply and clarification regarding Algorithm 2. I have several follow up questions.
> >
> > **Prolongation and restriction**
> >
> > Authors claim "our measurements show that even a single sparse matrix–vector multiplication at the problem scale (via cuSparse) is already slower than a full V-cycle of our method, making all methods that predicts sparse matrix-based prolongators noncompetitive in our problem settings."
> >
> > I kindly ask authors to explicitly answer the following questions:
> > 1. When restriction and prolongation operators are learned, they have the same structure as bilinear interpolation. If authors claim that multiplication by sparse matrix is too costly on GPU, how do they implement bilinear interpolation?
> > 2. Do authors claim that it is impossible to implement other operations with the same stencil as bilinear interpolation but with different weights efficiently on GPU?
> > 3. What kind of sparse matrix format authors used? Is it ELLPACK?
> >
> > **Details on equations**
> >
> > Still, authors do not provide enough details on equations:
> > 1. For the Helmholtz equation, please, specify how the wavenumbers were generated.
> > 2. PDE $-\Delta u - k^2 u = f$ is not a Helmholtz equation.
> > 3. Heat equations are still not specified in the appendix. Authors write " the diffusion coefficients are set to be constant inside the domain, but with smaller value near the bunny surface". What constant value precisely? Can the authors provide qualitative answers?
> > 4. The heat equation is not stationary. How is it discretised in time?
> >
> > I will provide an example of specification that I consider to be reasonably complete.
> >
> > *A would-be specification of linear system*
> >
> > We consider Poisson equation with source term
> > $$
> > -\frac{\partial^2 u(x, y)}{\partial x^2}-\frac{\partial^2 u(x, y)}{\partial y^2} - c(x, y) u(x, y) = f(x, y)
> > $$
> > on the square $(x, y) \in \Gamma = (0, 1)^2$ with Dirichlet boundary conditions $u(x, y) = 0$ for $(x, y)\in\partial\Gamma$ where $\partial\Gamma$ is the boundary of $\Gamma$.
> >
> > We use second order finite difference scheme on the uniform grid $x_i = i/N, x_j = j/N$ for $i, j = 1,\dots, N-1$ that for inner point reads
> > $$
> > \frac{-4u_{i,j} + \sum_{l=\pm1}\sum_{k=\pm1}u_{i+l,j+k}}{h^2} - c_{i,j}u_{i,j} = f_{i,j},
> > $$
> > where $h = 1/N$, $u_{i,j} = u(x_i, y_j)$ and likewise for $c$ and $f$.
> >
> > To generate dataset we select $c(x, y) = c_3(1 + \cos(c_1\pi x)\sin(c_2\pi y))^2$ where $c_1, c_2, c_3$ are generated from uniform distribution on $[1, 1000]$. In all experiments we select $f(x, y) = 1$.
> >
> > Can the authors provide the description of equations with this level of details?
> >
> > **Chebyshev smoothers**
> >
> > Authors seem to conflate implementation with the description of the algorithm. It is possible to write down GS method in the form of matvecs too, but authors decide somewhat arbitrarily that red-black GS method will be implemented efficiently and Chebyshev method is going to use matvec products. Can the authors clarify why they believe Chebyshev iteration is impossible to implement efficiently on CUDA?

---

> ### Author Response · Authors · 2025-12-03
> **Respond to the Follow-up Questions**
>
> We would like to thank the reviewer for the follow-up questions, and below we provide detailed clarifications.
> ## Our Strong Baseline Details
>
> We would like to clarify the background and the current implementation considerations of fast GPU-based grid Poisson solvers. Our GPU-based Poisson baseline solver design choices are already the state-of-the-art ones [Hu2021;Shao2022].
> A brief summary on such choices are:
>
> - We inherit a matrix-free strategy for both baseline and our method, the same  as in [Hu2021;Shao2022].
>
> - Under the same optimized tiling strategies of the matrix coefficients, the RBGS smoother is consistently at least $2\times$ faster than the Chebyshev polynomial smoother with the formulation in [Brannick 2013] (https://arxiv.org/pdf/1310.8385).
>
> - The performance gap grows when considering implementation-level overheads associated with spatially varying operators such as learned prolongators, therefore the acceleration on large-scale problems demonstrated in our manuscript is unprecedented and with nontrivial contribution.
>
> Below we provide detailed clarification on the reviewer's points.
> ### Matrix Format
> Our baseline solver does not use ELLPACK or any mainstream general-purpose sparse matrix format. Instead, we follow the "matrix-free" version of [Shao 2022] (see the remark in Appendix C.2), where matrices are stored implicitly through tiled coefficients within local cells. Under this formulation, both the matvec product and RBGS smoothing can be executed using costumized kernels, enabling substantial GPU acceleration compared to a conventional "matrix-based" sparse format.
>
> ### Restriction and Prolongation
>
> As shown in line `414-419` and line `454-455` of `model.py` in the supplementary code, the restriction and prolongation operators are implemented as a convolution with a **homogeneous**, compile-time-known stencil. This structure allows the operator to benefit from highly optimized CUDA acceleration. With such simple restriction and prolongation operators, the coarsened matrices can be generated using compile-time-known weighted pooling, which is as efficient as in prior GPU fluid solvers. We adopt these operators because they are used in SOTA stable fluid simulators [Hu 2021] and [Shao 2022], and our codebase is directly extended from theirs without modifying these settings.
>
> In contrast, prolongation-learning approaches employ **spatially varying kernels**. Their heterogeneous stencils incur more memory reads, resulting in both **precomputation overheads** (generating coarsened matrices) and **per-iteration** overheads.
>
> **The lack of publicly available implementations** of these learned prolongators prevents a fully controlled comparison. However, based on the performance of the current SOTA GPU implementations we rely on, substantial implementation-level overheads appear unavoidable for spatially varying prolongation operators under the computational constraints.
> ### Chebyshev Polynomial Smoothers
> The reviewer cites [Adams 2003] and [Brannick 2013] demonstrating that Chebyshev polynomial smoothing is a well-established component of classical geometric multigrid (GMG). We are fully aware of its historical importance in that context. In constrast, modern GPU-accelerated SOTA fluid pressure Poisson solvers employ a matrix-free UAAMG framework rather than classical GMG, and these systems consistently adopt RBGS smoothers [Hu 2021; Shao 2022]. While [Adams 2003] and [Brannick 2013] remain foundational, they **predate the SOTA GPU methods** that our work directly builds upon. Moreover, our experiments show that Chebyshev smoothing is at least $2\times$ slower as RBGS under this modern framework. As mentioned in our previous response, we have implemented both the second-order Chebyshev smoother and the RBGS smoother using the same matrix-coefficient tiling strategy as in [Hu 2021; Shao 2022], and the computational order of the Chebyshev smoother follows the specification in Algorithm 3.1 of [Brannick 2013]. This ensures that we have not "conflate implementation with the description of the algorithm", and provides a fair and balanced comparison.
> ## Time-Dependent PDE Settings
> The equations demonstrated in the work follow the standard practice established in FluidNet [ICML 2017], DCDM [ICML 2023], and MLPCG [ICML 2024]. These works primarily focus on the pressure Poisson equation arising in fluid dynamics. We also encompass other SPD linear systems induced by time-dependent PDEs, including diffusion equations for heat transfer and Helmholtz equations for wave propagation. More specific details are provided in the latest Appendix E.
> ### Clarification on Formula
> A standard Helmholtz equation takes the form $-\Delta u = \lambda u$ [p323, Evans 2010]. We consider the inhomogeneous variant $(-\Delta - \lambda)u = f$.
>
> ## Summary
> In summary, we believe that our test problem settings are standard and the comparisons of baselines are fair. We sincerely thank the reviewer for the careful evaluation.

---

### Official Review · Reviewer_bZug · 2025-10-31

**Soundness:** 3
**Presentation:** 4
**Contribution:** 3
**Rating:** 4
**Confidence:** 3

**Summary:**

The paper introduces a dual-channel neural multigrid preconditioner. This hybrid approach is designed around a dual-channel expression of the V-cycle. The authors propose to integrate a classical smoothing path with a lightweight, data-driven neural convolutional path. The proposed approach achieves a wall-clock time speedup of $1.03 - 1.26 \times$ over the GMG baseline for Poisson equations and a $2 - 3 \times$ acceleration on second-order PDEs (Heat and Helmholtz equations). It demonstrates a $5 - 10 \times$ improvement over MLPCG. The authors construct a large-scale benchmark (linear SPD systems up to $64$M unknowns) to fairly evaluate methods.

**Strengths:**

1. The authors propose a dual-channel neural multigrid preconditioner that reframes the standard Multigrid V-cycle to inject a lightweight neural convolutional path while strictly preserving the necessary SPD property.
2. The idea of rigorously benchmarking against a GMG solver is a significant original step in execution.
3. The experiments are conducted on a new large-scale dataset (linear SPD systems up to $64$M unknowns), demonstrating the method's ability to scale effectively to real-world scientific problems. The results are presented clearly, allowing for direct comparison to the strong GMG baselines.

**Weaknesses:**

1. The use of the CMA-ES optimizer is a significant practical drawback. The paper cites the reason: "challenging computational graphs for auto-differentiation tools," which suggests a fundamental design limitation. Gradient-free methods typically scale poorly to the high-dimensional parameter spaces common in deep learning, and they limit the ability to leverage standard, highly-optimized deep learning frameworks (e.g., PyTorch, TensorFlow).
2. The article does not answer the fundamental question: why is the computational graph intractable for standard backpropagation?
3. The performance gains on the second-order PDEs (Heat and Helmholtz equations), which feature more complex physics and operators, are less discussed in depth compared to the Poisson speedup. Helmholtz equations, in particular, are difficult for classical Multigrid due to their frequency content.

**Questions:**

See the Weaknesses

---

> ### Author Response · Authors · 2025-11-23
>
> We thank the reviewer for the insightful comments and the opportunity to clarify our design choices and experimental scope. Below, we provide detailed responses to the concerns regarding CMA-ES training and more discussion on the heat and Helmholtz equations.
>
> ## CMA-ES Training
>
> We address the reviewer’s concern regarding our use of CMA-ES instead of backpropagation-based training. The core challenge lies not in the algorithmic suitability of gradient descent, but in the **practical limitations of existing auto-differentiation (AD) frameworks when applied to large-scale multigrid preconditioners with irregular data structures**.
>
> We have attempted two major AD frameworks:
>
> - **Taichi** – While Taichi is specifically designed for dynamic, irregular data structures with SOTA performance [Hu2019, Hu2021], applying its AD to our full preconditioner resulted in compilation failure due to the enormous number of SNodes produced by the underlying computational graph.
> - **PyTorch** – A sparse-matrix-based PyTorch prototype led to out-of-memory errors, suggesting that certain operations silently fall back to dense kernels internally. This makes it unsuitable for our large, structured-but-sparse computation.
>
> These observations indicate that supporting gradient-based optimization for our problem requires **substantial custom engineering**, such as designing tailor-made CUDA kernels and custom gradient rules. While such engineering is valuable, it is beyond the intended scope of this work, which focuses on algorithmic effectiveness at scale rather than on building a new AD system.
>
> Additionally, for our preconditioner design, the number of learnable parameters is relatively small (~1K). In this regime, **gradient-free methods are often more efficient and robust**. Recent works such as PCGBandit [1] demonstrate that lightweight, derivative-free optimization can outperform strong numerical baselines at scales, performance not achieved by earlier gradient-based methods. Gradient descent becomes more advantageous primarily when model sizes are large (e.g., the 32K-parameter MLPCG variant), or when leveraging deep-learning-scale capacity.
>
> Given these considerations, CMA-ES offers a practical, stable, and scalable solution within our problem constraints. Investigating fully scalable gradient-based training—together with the required low-level systems support—remains an interesting direction for future research.
>
>
>
> ## Additional Discussions on Heat and Helmholtz Equations
>
> Our work continues the research line of neural conjugate-gradient solvers, as seen in FluidNet (ICML 2017), DCDM (ICML 2023), and MLPCG (ICML 2024). As is common in this literature, these methods are primarily evaluated on fluid-related Poisson equations. To better illustrate the **applicability boundary** of neural CG methods, we additionally include spatially varying Poisson problems, heat diffusion, and an SPD subset of Helmholtz equations.
>
> We have included detailed parameter specifications for all PDE families in Appendix E and provide extended evaluation results in Appendix I in the revised version.
>
> ## Summary
>
> We sincerely appreciate the reviewer's questions, which help clarify the scope and motivation of our methodology. Our goal is to highlight the potential and limitations of neural CG-based preconditioners across representative PDE families, and close the performance gap by the proposed neural preconditioner. We hope the expanded explanations address the reviewer’s concerns, and are happy to elaborate on any further questions.
>
>
> [1] Khodak, M., Jung, M. K., Wynne, B., Chow, E., & Kolemen, E. (2025). PCGBandit: One-shot acceleration of transient PDE solvers via online-learned preconditioners. arXiv preprint arXiv:2509.08765.

---

### Official Review · Reviewer_Pbu3 · 2025-11-01

**Soundness:** 4
**Presentation:** 4
**Contribution:** 2
**Rating:** 4
**Confidence:** 4

**Summary:**

The paper proposes a hybrid multigrid preconditioner for PCG that runs a lightweight learned path (H) alongside a classical smoothing path (S). Using an equivalent V-cycle formulation, the authors argue the overall preconditioner remains SPD, preserving PCG guarantees. Training minimizes the k-step CG residual (self-supervised) with a derivative-free optimizer. On large regular-grid Poisson/Heat/(SPD) Helmholtz problems, the method reports GMG-like per-iteration cost with fewer iterations, outperforming prior neural preconditioners.

**Strengths:**

Consistent preservation of SPD guarantees.
By formulating the preconditioner as an equivalent V-cycle with parallel S/H paths, the overall operator is kept symmetric positive definite, thereby preserving PCG’s convergence guarantees while introducing learnable components. This is a practically critical property in this domain.

Well-tuned GMG as the primary baseline.
The comparisons are made against a carefully engineered geometric multigrid (GMG) rather than a textbook/default setup, so the results are meaningful regardless of win/loss and the risk of overstating gains is reduced.

Self-supervised objective (no labels required). By optimizing the k-step residual as the training objective, the method avoids ground-truth solutions and other costly supervision. This is practical for physics applications where data preparation is often the bottleneck.

Classical S-path as a safety net (robustness foundation). When the learned H-path underperforms out of distribution, the classical MG S-path still ensures baseline convergence. This delivers a clear risk profile—learning provides upside; classical MG guarantees the floor—which is appealing for real deployments.

**Weaknesses:**

1. Scope limitation. Experiments focus on regular-grid SPD elliptic problems (Poisson/Heat/(SPD) Helmholtz) whose spectra are closely related; effectiveness is shown within this family, not beyond it.
2. Out-of-distribution (zero-shot) is thin. Systematic tests where RHS/coefficients/BCs deviate substantially from the training distribution are limited. Results on an unseen scene (e.g., Worm) remain in-family and do not establish cross-family generalization.
3.Boundary-dominated settings (still SPD) are untested. No evaluation with obstacles/rotating bodies/cut-cell or immersed-boundary style setups where irregular boundary stencils dominate; robustness near boundaries is unclear.
4. Resolution generalization is limited. There is an extrapolation 256³→512³ for Poisson, but no systematic sweeps for Heat/(SPD) Helmholtz, nor train@low → test@high cross-resolution tests.
5. Comparison protocol gap. Prior work shows “train without objects → zero-shot to scenes with objects.” In contrast, this paper does not provide a comparable boundary-dominated protocol, weakening claims of geometric/BC diversity.

**Questions:**

1. Training cost & amortization. Please report CMA-ES trial counts, per-trial wall-clock, hardware, and an estimate of train-once-solve-many payback.
2. Boundary-dominated (SPD) tests. On obstacle Poisson and rotating-body Poisson (3D 256³), can you report iterations, wall-clock, peak memory, boundary-zone (≤2 cells) residual decay, and preconditioned condition number?
3. Cross-resolution. What are the degradation ratios for train@256³ → test@512³/384³ (iterations, time, condition number)? Is retraining necessary?

---

> ### Author Response · Authors · 2025-11-23
>
> We the reviewer for their valuable comment. Below, we address the questions and concerns in detail.
>
> ## Cross-Resolution Generalization
>
> We have added additional resolution-generalization experiments in the revised version. The updated results are summarized below:
>
> - The model trained on 50 Poisson instances generalizes to the 4K ShapeNet airplane geometries with an average **1.1×** speedup.
> - Strong acceleration persists in cross-resolution evaluation at \(384^3\), achieving up to **4×** speedup **without any retraining**.
>
> The main paper already demonstrates broad generalization across diverse boundary geometries. Our test set includes variations in object translation, rotation, and deformation (see videos), representing significantly richer diversity than prior works such as DCDM and MLPCG. The new experiments further confirm that our method delivers effective acceleration under extensive distribution shifts.
>
> ## Problem Scope
>
> Our work advances neural conjugate-gradient–based solvers on voxel grids, continuing the established line of research in FluidNet [ICML 2017], DCDM [ICML 2023], and MLPCG [ICML 2024]. These works consistently evaluate neural CG methods on fluid Poisson equations.
>
> To more comprehensively assess the applicability of neural CG approaches, we extend this scope to include spatially varying Poisson equations, heat diffusion, and SPD Helmholtz equations. These additional experiments delineate the practical boundary within which neural CG–based preconditioners remain effective.
>
> ## Training Without Objects and Zero-shot Generalization
>
> DCDM [ICML 2023] shows that training exclusively on square domains provides limited generalization even for uniform boundary conditions. MLPCG [ICML 2024] further demonstrates that such training fails under mixed boundary conditions. We would like to stress that this form of zero-shot transfer is notably poor in performance, even not competitive for weak numerical baselines.
>
> The limitation is structural: training only on a square domain cannot expose the model to **locally varying operator kernels** arising from mixed Dirichlet–Neumann boundaries. These heterogeneous row structures cannot be inferred from a single homogeneous training setup, making zero-shot generalization fundamentally unreliable.
>
> ## Boundary-Cell Residual Decay
>
> We added additional tests evaluating boundary-cell residual decay using the first matrix released in the Heat Bunny scene. Below we report the CG relative residual on all cells two grid cells away from the boundary:
>
> | Iter | 1 | 2 | 3 | 4 | 5 | 6 | 7 |
> |------|---|---|---|---|---|---|---|
> | GMG | 7.7e-01 | 5.0e-01 | 3.3e-01 | 2.2e-01 | 1.2e-01 | 7.4e-02 | 4.2e-02 |
> | Our | 7.7e-01 | 1.2e-01 | 2.6e-03 | 1.8e-04 | 2.1e-05 | 2.0e-06 | 3.1e-07 |
>
> These results confirm stable decay behavior near complex boundary regions.
>
> ## Training and Testing Cost Balance
>
> The discussion of “train-once-solve-many payback" is typically not addressed in the line of literature. However, we notice that since it is demonstrated, that the model trained on 50 instances generalized to 4K scenes (potentially with hundreds of instances in each scene), it is possible to "amortize" the training cost, which is unprecedented for works in this line.
>
> | Method   | Instances Required to Amortize Training Cost (vs. GMG) | Per Solving Improvements (vs. GMG) | Training Cost Per Instance
> |----------|---------------------------------------------------------|--|---|
> | **Ours: Poisson** | ~1,800,000                                                | ~20ms | 1h |
> | **Ours: Heat** | ~7,200                                                | ~100ms | 12 min |
> | **MLPCG** | $\infty$                                             | <0 | > 2h* |
> | **DCDM**  | $\infty$                                             | <0 | >2h* |
>
>
> \* Obtaining Ritz vectors for these works takes significantly additional time cost.
>
> Among existing neural preconditioners, **ours is the only method whose training cost can be amortized realistically in large-scale applications**. This constitutes the **first practical demonstration** of a neural CG-based preconditioner whose end-to-end computational cost is competitive with state-of-the-art numerical solvers. We hope this motivates further exploration of neural solvers with practical real-world cost profiles.
>
>
> ## Summary
>
> We have expanded our resolution-generalization evaluations, introduced more diverse geometric tests, clarified the broader problem scope beyond standard Poisson setups, and explained the fundamental limitations of training solely on square domains. We hope that these clarifications fully address the reviewer’s concerns and welcome any further questions.

---

### Author Response · Authors · 2025-11-23
**General Clarification**

We thank all reviewers for their valuable and constructive feedback. We have revised the manuscript incorporating additional experiments on generalizability and providing more detailed descriptions of the training setup. Below, we address some concerns raised during the review process.

## Novelty and Contribution

Our work focuses on the recent advancement in neural conjugate gradient methods targeting typically on Poisson problems in fluid animation, a setting established by the previous work FluidNet [ICML2017], DCDM [ICML2023], and MLPCG [ICML2024]. We additionally provide some heat diffusion equations and a subset of SPD Helmholtz equations to investigate the application boundaries of the neural CG methods.

To the best of our knowledge, no existing neural methods can accelerate Poisson solving for fluid animation at practical scales (e.g., $256^3$) while maintaining accuracy guarantees. State-of-the-art solutions in this domain remain MGPCG-based methods [Hu2021, Shao2022], while recent advances in neural multigrid have not yet been integrated into high-performance fluid animation pipelines. Our work highlights this gap: current neural approaches fall significantly short of well-optimized numerical baselines in this setting.

Another long line of literature explores neural multigrid methods, but the majority targets small- to moderate-scale problems. Due to the lack of open-source implementations, direct comparison is limited. However, our analysis and experiments indicate that prolongator-prediction approaches are unlikely to achieve state-of-the-art performance at large resolutions on regular grids, as their inference overhead and per-iteration costs on applying sparse matrix-based prolongations become less competitive compared to the GMG baseline with RBGS smoothing and rule-based Galerkin coarsening on grids.

In this context, our aim is to address the large-scale challenge, where practical constraints render existing neural methods insufficient. Our work provides an initial step toward a neural solution that remains competitive under real computational limits.

## Generalizability

To further examine cross-scene and out-of-distribution generalizability, we conducted an additional experiment using all 4,045 voxelized airplane models from ShapeNet as obstacles, as described in the newly added Appendix I. These models possess geometries and topologies distinct from the training domains. The method generalizes effectively to these unseen boundary configurations, achieving improvements of 1–4 CG iterations in most cases and an average **wall-clock acceleration of about 10\%**.

## Condition Number Improvements

For our largest systems (16-64M DoFs), explicitly computing the condition number is impractical on typical workstations or lab-scale hardware due to the cubic complexity of eigenvalue decomposition. Moreover, obtaining the explicit matrix form of the neural MG preconditioner is difficult, as it is sparse in computation but dense when fully materialized. Given these limitations, and since the condition number is only an indirect indicator of solver quality, we focus instead on the more relevant metrics of wall-clock time and iteration performance.


## Summary

In summary, our work addresses a practically important yet previously underexplored challenge: developing neural conjugate gradient methods achieving SOTA at the real-world simulation scales. We demonstrate meaningful improvements over established baselines **(\$5\times\$ faster than previous neural SOTA)**, strong cross-scene generalizability, thereby providing a step for future research on scalable neural PCG methods.

---

### Author Response · Authors · 2025-12-03
**Summary of the Rebuttal Phase**

Dear Area Chairs,

Thank you for your hard work during the reviewing process. We respectfully provide a concise summary addressing the main concerns raised in the reviews, and we highlight the core contributions of our submission.

## Novelty and Contributions

Our work builds on the well-established line of neural solvers for fluid Poisson solving: FluidNet [ICML 2017], DCDM [ICML 2023], and MLPCG [ICML 2024]. Our contributions are threefold:

1. **Comprehensive evaluation** revealing that the state-of-the-art (SOTA) numerical methods remain faster than neural SOTA by an order of magnitude on large-scale voxel-grid Poisson equations.
2. Introduction of a **hybrid dual-channel neural preconditioner** that guarantees SPD structure.
3. Advancement in performance: compared to the strong GMG baseline, our proposed method obtains a 1.03-1.26$\times$ speedup on Poisson equations and 2-3$\times$ acceleration on other second-order PDEs involving up to 64 million unknowns, while also delivering 5-10$\times$ improvements over the SOTA neural method. These results establish a new benchmark for neural conjugate gradient research on large-scale voxel-grid problems.

##  Generalizability Across Geometries and Resolutions

 Reviewer Pbu3 has expressed concerns about how well our method generalizes to diverse geometries and cross-resolution evaluation. In the revised manuscript, we added new experiments. The model trained on only 50 Poisson instances generalizes to **4K ShapeNet airplane geometries** with an average **1.1$\times$ speedup** on the fluid Poisson equations, and maintains **up to $4\times$ acceleration at $384^3$ resolution without any retraining** on the heat and Helmholtz equations . The main paper already includes highly diverse boundary geometries including translation, rotation, and deformation, which exceed the variability tested in prior works such as DCDM [ICML 2023] and MLPCG [ICML 2024].

## Fairness of Baselines
There were also questions from Reviewer gCMy regarding the fairness of our numerical and neural baselines. We summarize that our GPU-based Poisson solver baseline is fully aligned with the latest **SOTA matrix-free implementations** used in [Hu 2021] and [Shao 2022]. Other numerical baselines, such as [Brannick 2013] mentioned by the reviewer, are less competitive than mordern baselines we choose. For other weak neural baselines, the missing of publicly available code prevents from controlled comparisons, but our experiments show that the spatially varying learned prolongation operators introduce substantial implementation overhead when scaling up to the demonstrated problem scale. We hope this clarifies why our chosen baselines are both standard and appropriate for the problem setting.



We sincerely appreciate the reviewers' feedback and your hard work in the review process. We hope the above clarifications help consolidate our responses and assist in your final assessment.

Thank you very much for your time and consideration.



Sincerely,

The Authors

---

### Meta-Review · Area_Chair_xixz · 2026-01-06

**Summary:**

This submission studies learned preconditioning for large-scale PCG on regular 3D grids and is motivated by a gap between “neural MG / neural CG” papers that look strong at small scales and the reality that production fluid simulation still rely on highly optimized MGPCG. The paper’s central proposal is a dual-channel, V-cycle-style preconditioner that is intended to preserve the symmetry structure required by PCG. Empirically, the authors focus on structure grid SPD systems (Poisson, diffusion/heat, and an SPD “Helmholtz subset”, at scales up to tens of millions of unknowns. On Poisson, the reported gains over a tuned geometric multigrid baseline are modest (roughly 1.03–1.26× depending on the scene), while the paper reports larger improvements on the other PDE families and much larger improvements over prior neural baselines such as MLPCG.

**Reviewer Concerns:**

The reviewers agree that the paper takes the baseline question seriously that the authors implement and benchmark against a highly optimized matrix-free geometric MGPCG baseline, and they also include AMGX/AMGCL comparisons. Reviewers also appreciated the fact that the experiments are conducted at realistic problem sizes. The overall presentation are generally viewed as solid.

At the same time, the reviewers’ common concerns remain substantive, and in my view they are only partially addressed by the author responses and the limited discussion. The recurring issues are (i) novelty/positioning, (ii) the breadth and rigor of the generalization story, (iii) the framing of the “positive definiteness” in the manuscript.

On novelty, the most skeptical reviewer gCMy argues (reasonably) that “learning or optimizing multigrid components” is a well-established line of work, and that the paper’s claim should not be “first neural preconditioner to surpass optimized MG” in a general sense. The authors’ response is essentially that the relevant comparator class is not “any learned MG paper,” however, this narrowing is not fully justified as written. In particular, a direct comparison to recent neural multigrid-style methods (e.g., Xie et al., 2025) seems necessary, given the conceptual and architectural similarity (a V-cycle framework with a learned smoother). I believe Xie et al. also claim their methods outperform GMG.

On generalization, the most consistent criticism is that the evaluation envelope is narrow: regular grids and SPD elliptic-like operators whose spectra are closely related. Even for these tasks, the gains over GMG are not siganificant. The rebuttal adds additional experiments, including a large-scale ShapeNet-style obstacle set and some cross-resolution evaluation, which helps materially. It strengthens the paper, but it does not fully address the reviewer’s point that the generalization story is still limited to a fairly tight problem family.

Finally, an important point in the discussion is that while the preconditioner is symmetric by construction, the “positive definiteness” aspect is not enforced as a hard constraint during training. The authors explicitly describe the method as “empirically SPD”. This should be stated very clearly and precisely in the main paper, and it weakens the strength of the theoretical framing as currently written.

**Reviewer Scores:**

I do not expect major movement. The positive reviewer indicated they would keep their score after clarification. The novelty-focused reviewer gCMy raised follow-ups and still appears unconvinced on the key positioning point, so I would expect that score to remain a 4. The reviewer most concerned about generalization got some additional experiments in the rebuttal, but the remaining limitations are still real; I would expect that score to remain around 4. The reviewer bZug and Pbu3 most focused on training might move slightly upward if fully satisfied by the training justification, but given they also flagged limited discussion of the harder PDE cases, I would still expect them to remain at 4 or, at best, a 5.

---

### Decision · Program_Chairs · 2026-01-26

Reject